

# Sensitivity of global direct aerosol radiative forcing to uncertainties in aerosol optical properties

Jonathan Elsey[1], Nicolas Bellouin[1], and Claire Ryder[1]

[1]Department of Meteorology, University of Reading, Reading, RG6 6BB, UK

*Correspondence to*: Claire Ryder (c.l.ryder@reading.ac.uk)

## Abstract

New satellite missions promise global reductions in the uncertainties of aerosol optical properties but it is unclear how those reductions will propagate to uncertainties in the shortwave direct aerosol radiative effect (DARE) and radiative forcing (DARF), which are currently large, on the order of at least 20%. In this work we build a Monte-Carlo framework to calculate

the impact of uncertainties in aerosol optical depth (AOD), single scattering albedo (SSA) and asymmetry parameter on the uncertainty in shortwave DARE and DARF. This framework uses the results of over 2.3 million radiative transfer simulations to calculate global clear-sky DARE and DARF based on a range of aerosol optical property uncertainties, representative of existing and future global observing systems. We find the one-sigma uncertainty varies between $\pm0.23$ to $\pm1.91$ Wm$^{-2}$ (5 and 42%) for the top of atmosphere (TOA) clear-sky DARE and between $\pm0.08$ to $\pm0.47$ Wm$^{-2}$ (9 and 52%) for the TOA DARF.

At the TOA, AOD uncertainty is the main contributor to overall uncertainty, except over bright surfaces where SSA uncertainty contributes most. We apply regionally varying uncertainties to represent current measurement uncertainties, finding that aerosol optical property uncertainties represent 24% of TOA DARE and DARF. Reducing regionally varying optical property uncertainties by a factor of two would reduce their contributions to TOA DARE and DARF uncertainty proportionally. Scaling to all-sky conditions, aerosol optical property uncertainty contributes to about 25% total uncertainty in TOA, all-sky DARE

and DARF. Compared to previous studies which considered uncertainties in non-aerosol variables, our results suggest that the aerosol optical property uncertainty accounts for a third to a half of total uncertainty. Recent and future progress in constraining aerosol optical properties using ground-based or satellite retrievals could be translated into DARE and DARF uncertainty using our freely available framework.

## 1. Introduction

Aerosols are one of the major contributors to the radiative forcing of Earth's climate via changes in its radiation budget. In addition to their indirect effects on climate due to their influence on cloud microphysical properties, aerosols also interact with radiation directly via absorption and scattering. The effect on the radiation budget due to these aerosol-radiation interactions is referred to as the "direct aerosol radiative effect" (DARE, also called radiative effect of aerosol-radiation interactions in IPCC assessment reports), while the effect of the change in aerosol distributions from pre-industrial times due to only



anthropogenic aerosols is termed the "direct aerosol radiative forcing" (DARF, also called radiative forcing of aerosol-radiation interactions). These quantities are typically considered at the surface and at the top of atmosphere (TOA).

Quantification of the magnitude of the aerosol radiative forcing is a major challenge that has motivated a significant body of research over the last 30 years (Bellouin et al. (2020)). Although uncertainties in effective radiative forcing are dominated by

aerosol-cloud interactions, and has been the focus of much recent work, uncertainties due to direct radiative effects are still large, and on the order of 100% (Forster et al., 2021). Aerosol-radiation and aerosol-cloud interactions depend on different aerosol properties and atmospheric processes. In this study we focus on direct aerosol-radiation interactions, and the aerosol properties relevant to them.

There have been recent attempts to constrain the value of DARE and DARF, using a variety of methodologies. Bellouin et al. (2013) used satellite data from MODIS assimilated into the MACC aerosol reanalysis to estimate clear-sky (cloud free) and all-sky TOA/surface DARF and DARE. Kinne (2019b) used a two-stream radiative transfer code with 8 SW bands, in conjunction with the Max-Planck Aerosol Climatology version 2 (MACv2, Kinne (2019a), see also Section 2.1), to obtain estimates of clear and all-sky TOA and surface DARE and DARF, separated into SW and longwave (LW) components, as

well as by aerosol type. An uncertainty estimate was obtained for the total aerosol radiative forcing but was not separated into uncertainties for the direct and indirect effects separately. Matus et al. (2019) obtained vertical profiles of clouds and aerosols from CloudSat and CALIPSO observations and used radiative kernels to estimate clear and all-sky DARE and DARF at the TOA. Thorsen et al. (2020, 2021) applied radiative kernels derived using MERRA-2 observations to estimate TOA DARE and its uncertainties (Thorsen et al., 2020), and then obtain similar kernels for TOA DARF (Thorsen et al., 2021). This

approach allows for systematic estimation of the uncertainties, particularly due to aerosol optical properties. These optical properties are based on those obtained via the AERosol Robotic NETwork (AERONET; Holben et al., 1998), using a matching algorithm to pair AERONET sites with similar aerosol characteristics to gridded MERRA-2 reanalysis data. This gives a "best estimate" uncertainty based on a hypothetical global observing system with AERONET-like accuracy. They also provide a hypothetical "enhanced" estimate of this uncertainty, by assuming the single scattering albedo of highly scattering aerosols is

known perfectly (equal to 1 in the visible spectrum), and that it is only directly retrieved for absorbing aerosols, in addition to assumed improvements in vertical profiles via lidar measurements. The results of those studies are summarised in Table 1 for clear-sky conditions, which is the primary focus of the present work. They suggest that these different approaches generally agree on the central value of DARE and DARF. There remains however a large relative uncertainty in DARE and DARF across different studies, typically on the order of at least ~20% or greater.


There are many factors which control uncertainties in DARE and DARF. These include uncertainties in aerosol loading, optical properties, and anthropogenic fraction, as well as biases inherent to the aerosol environment, such as cloud properties, surface albedo, and gaseous absorption (Stier et al., 2013). Radiative transfer considerations, such as the spectral resolution used in





the calculations, also play a role (Randles et al., 2013). Parameters relating to aerosol distribution and optical properties are
typically measured and provided to the community via observations from satellites such as the Moderate Resolution Imaging
Spectroradiometer (MODIS), or via ground-based remote sensing observation networks, notably AERONET. The ongoing
Metrology for Aerosol oPtical Properties (MAPP) project aims to make significant reductions in the uncertainties of retrievals
of aerosol optical properties, in particular using the Generalized Retrieval of Aerosol and Surface Properties (GRASP)
algorithm (Dubovik et al., 2021; Herrera et al., 2022). Next-generation satellite retrievals of aerosol optical properties are also
expected, such as EarthCare (Wehr et al., 2006) and Plankton, Aerosol, Cloud, Ocean Ecosystem (PACE; Werdell et al., 2019).
It is well known that Aerosol Optical Depth (AOD) and aerosol Single-Scattering Albedo (SSA) are primary drivers of
observation-based DARF uncertainties (e.g., Loeb and Su, 2010). However, the impact of increased accuracy and precision
of measurements and retrievals of the aerosol optical properties on DARE and DARF uncertainties has been less studied. In
the present study, particular attention is given to the role of uncertainties in the preindustrial reference state.


| Study | TOA | | | Surface | | |
|---|---|---|---|---|---|---|
| | DARE (W m$^{-2}$) | DARF (W m$^{-2}$) | Forc. eff. | DARE (W m$^{-2}$) | DARF (W m$^{-2}$) | Forc. eff. |
| Bellouin et al. (2013) | −7.3 ± 1.3 | −2.5 ± 0.5 | −41 | −10.8 ± 1.9 | −5.5 ± 1.0 | −60 |
| Kinne (2019b) | −3.5 | −0.69 | −33 | −7.4 | −1.9 | −58 |
| Thorsen et al. (2021) | −3.17 ± 0.85 | −0.67 ± 0.24 | n/a | n/a | n/a | n/a |
| Thorsen et al. (2021), enhanced | −3.17 ± 0.54 | −0.67 ± 0.16 | n/a | n/a | n/a | n/a |
| Matus et al. (2019) | −2.62 ± 0.6 | −0.77 ± 0.3 | n/a | n/a | n/a | n/a |
| This work, regionally varying uncertainties | −4.55 ± 1.09 | −0.93 ± 0.22 | −41.32 | −8.3 ± 1.97 | −2.1 ± 0.49 | −65.95 |
| This work (upper limit), globally uniform uncertainties | −4.55 ± 1.91 | −0.93 ± 0.47 | −41.32 | −8.3 ± 3.35 | −2.1 ± 0.92 | −65.95 |
| This work (lower limit) globally uniform uncertainties | −4.55 ± 0.23 | −0.93 ± 0.08 | −41.32 | −8.3 ± 0.37 | −2.1 ± 0.12 | −65.95 |

**Table 1: Top-of-atmosphere (TOA) and surface clear-sky Direct Aerosol Radiative Effect (DARE) and Direct Aerosol Radiative Forcing (DARF), both in W m$^{-2}$, for previous studies and this work, along with their uncertainties (where applicable). Thorsen et al. (2021), enhanced refers to the uncertainties in their "enhanced" methodology (see Section 2). The numbers for this work are reflective of the AERONET v1-like uncertainties (see Section 3), or the upper and lower limits of our sampled uncertainty range**
**(Table 3). Forc. eff. refers to forcing efficiency in W m$^{-2}$ per unit AOD.**



## 2. Methodology

### 2.1. Radiative transfer model setup

Radiative transfer calculations are performed with the 'UK Met Office Suite of Community Radiative Transfer Codes based on Edwards and Slingo' (SOCRATES) in its two-stream, 6-band shortwave configuration as used in the GA9 configuration of

the UK Met Office Unified Model (denoted in SOCRATES as sp_sw_ga9, updated from the GA7 configuration of Walters et al. (2019)). This configuration uses solar spectral irradiance from Lean et al. (2005), with gaseous absorption computed using the correlated-k distribution method with HITRAN 2012 spectroscopic data (Rothman et al., 2013) and what is referred to within SOCRATES as the Elsey-Shine water vapour continuum (see Elsey et al., 2020; Anisman et al., 2022).

Aerosols are prescribed using the MACv2 aerosol climatology (Kinne, 2019a). MACv2 provides AOD, SSA (denoted $\omega_0$), and $g$ for each month of the year for both present-day and pre-industrial cases for different aerosol types. MACv2 obtains these distributions with a combination of observations from the ground-based sun-photometer network AERONET and global aerosol modelling derived mostly from AeroCom Phase1 simulations (Kinne et al., 2006). We interpolate these properties to a 5°x5° latitude-longitude grid from the native 1°x1°, with 20 vertical levels. Calculations are done here on a seasonal average

to reduce the number of radiative transfer calculations by a factor of 3 with a limited impact on the calculated DARE and DARF. Aerosols are separated into anthropogenic fine mode, natural fine mode, and coarse mode aerosols. MACv2 provides gridded vertical profile information for AOD at 550 nm for fine-mode and coarse-mode aerosols, as well as spectral AOD, asymmetry factor and SSA for each type. These vertical profiles are combined with the spectral information and applied to the relevant aerosol types to obtain the vertically resolved AOD at each wavelength, scaling them proportionally to the AOD at

550 nm. To include the MACv2 aerosol optical properties in SOCRATES, it is necessary to transform the original AOD and SSA distributions into absorption and scattering coefficients at each gridpoint and vertical level. This is done by multiplying the vertically resolved AOD by the thickness of each vertical layer, as defined by MACv2, to obtain an extinction coefficient, and then multiplying this by $\omega_0$ or $1 - \omega_0$ to get scattering and absorption coefficients, respectively. This is done for each of the points in our 5°x5° latitude-longitude grid, for each aerosol type. The original MACv2 vertical profiles do not contain

information about single scattering albedo or asymmetry factor. These are therefore kept constant throughout the whole vertical profile. These resulting optical properties for the 16 SW spectral bands of MACv2 are then interpolated to the 6 bands used here in SOCRATES. These profiles are then perturbed depending on the relevant uncertainties, and then combined to create a single aerosol column.

Surface albedo is taken from the SOCRATES ocean albedo scheme over ocean, which accounts for the effect of changes in solar zenith angle, and satellite data from the Scanning Imaging Absorption spectroMeter for Atmospheric CHartographY version 2.6 (SCIAMACHY; Tilstra et al., 2017) over land, interpolated from the native 33 nm spectral resolution to the 6 SW



bands used in SOCRATES, and regridded to a 5°x5° latitude-longitude grid. More details on the surface albedo used can be found in Section 2 and the Supporting Information to Byrom and Shine (2022).


Standard atmospheric profiles (McClatchey et al., 1972) corresponding to latitude and time of year are used as the underlying climatology in 30° latitude bands. All simulations were performed in clear skies only. For each simulation, the radiative transfer code is called 3 times with different solar zenith angles computed according to the latitude and time of year, and the outputs combined using Gaussian quadrature to obtain the diurnally averaged irradiances.


## 2.2. Benchmark estimates of DARE and DARF

The unperturbed aerosol optical properties from MACv2 are used to calculate SW radiative fluxes at the top-of-atmosphere
and surface. The difference with a no-aerosol calculation provides DARE, while the difference with the pre-industrial calculation the DARF. Figure 1 shows the annual-mean reference TOA (top panel) and surface (lower panel) DARE, while Figure 2 shows the same for DARF.

Since the aerosol properties used here are derived from MACv2, also used in Kinne (2019b), these results can be directly
compared since the only differences are the methodological and modelling approaches. Both the upper and lower panels of Figure 1 show similar spatial distributions to the anthropogenic annual-mean clear-sky DARF shown in Figure 7 of Kinne (2019b) although in both cases the estimate from this work is about 0.2 W m$^{-2}$ larger. This is likely due to a combination of the various host model uncertainties detailed in Stier et al. (2013) and Randles et al. (2012).



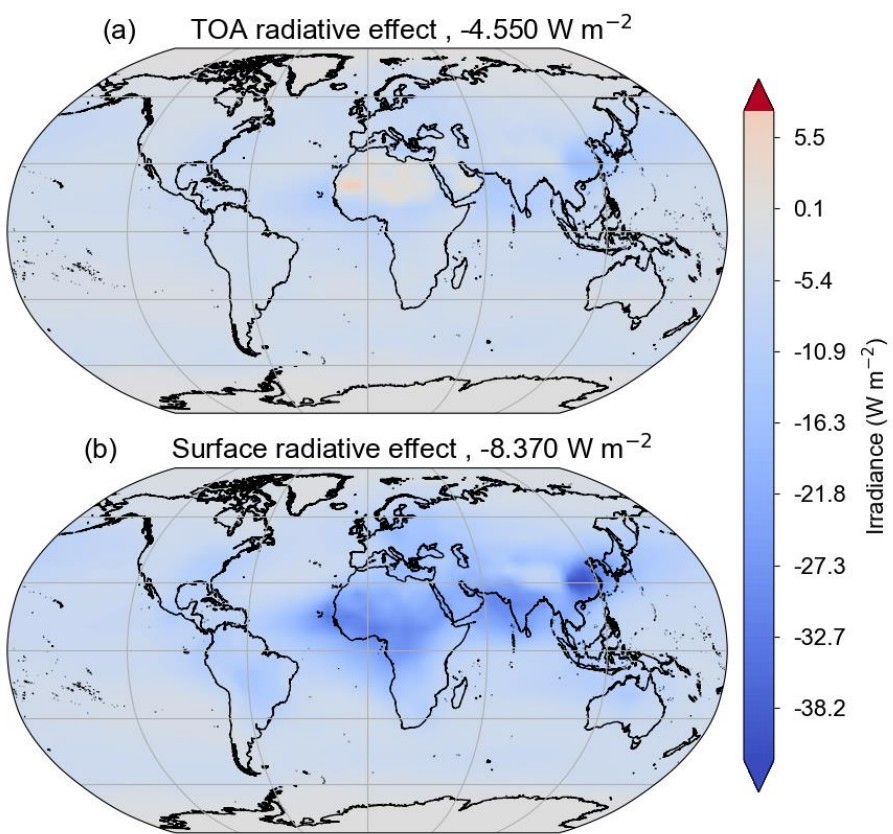

**Figure 1: Direct Aerosol Radiative Effect (DARE), in W m⁻² at the top of the atmosphere (TOA, panel a) and surface (panel b), as estimated using the SOCRATES radiative transfer code applied to the MACv2 aerosol climatogy. Global average values are given above each panel.**



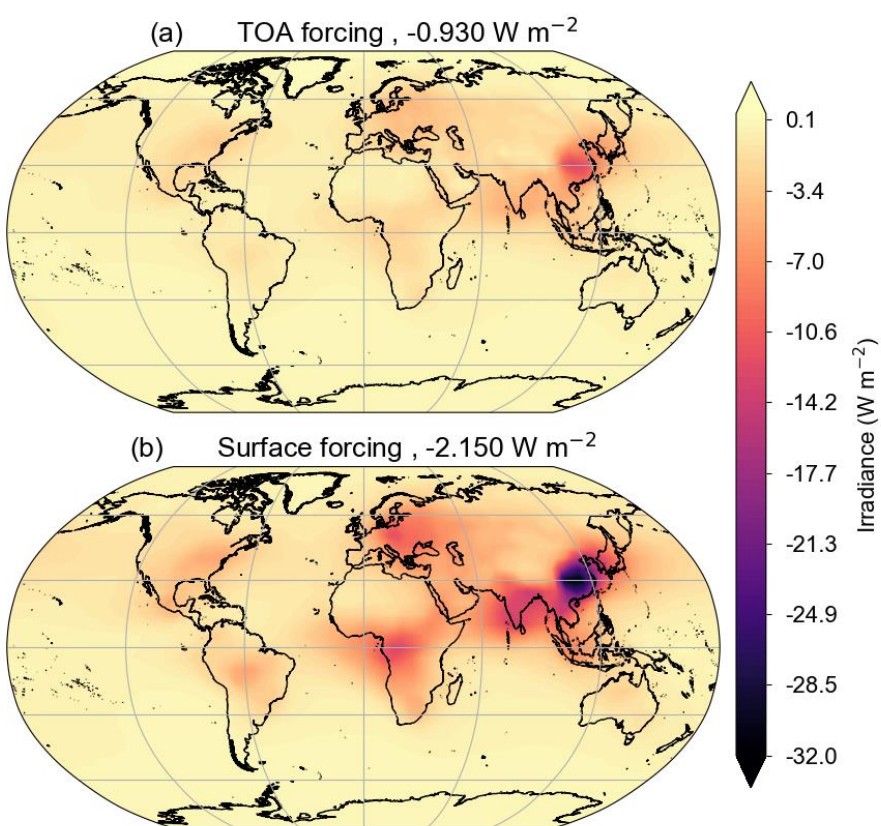

**Figure 2: As Figure 1, but for the Direct Aerosol Radiative Forcing (DARF), in W m$^{-2}$.**


The most significant differences are likely due to different assumptions about the surface albedo, the lower-resolution latitude-longitude grid and the coarser spectral resolution used in this work. To investigate the latter, an additional reference calculation was performed using the 260-band version of SOCRATES (referred to within the code as sp_sw_260_jm3), with the corresponding interpolation of surface and aerosol properties. The differences between the 6 and 260-band versions of the DARF calculation are shown in Figure 3. While there are biases of up to 5% spatially, these almost entirely cancel out when averaging over the globe for both the TOA and surface. This results in a more negative DARF by 1 to 2% at both the TOA and surface for the 260-band case, further increasing the differences between this work and Kinne (2019b), which used 8 solar wavebands. While the spatial differences will likely result in biases when calculating uncertainties (see Section 3), these are






also likely to average out on a global-mean scale and therefore the decreased spectral resolution should not significantly impact those results.

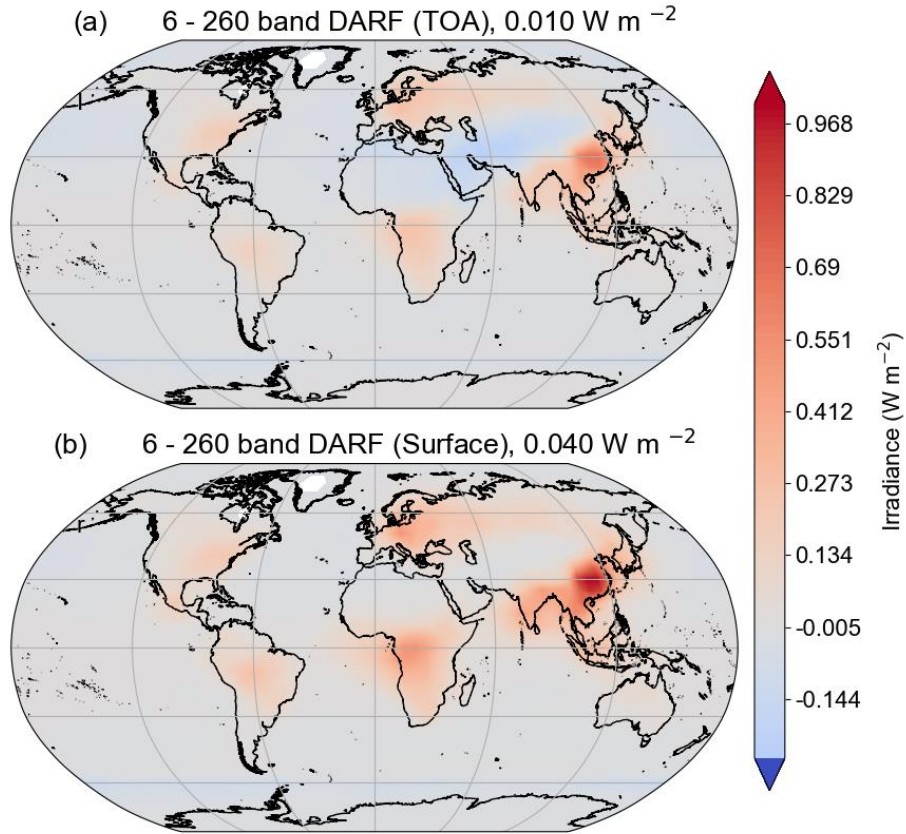

**Figure 3: Differences in top-of-atmosphere (TOA, upper panel) and surface (lower panel) Direct Aerosol Radiative Forcing (DARF), in W m⁻², between radiative transfer calculations using a 260-band version of SOCRATES, with corresponding higher resolution aerosol optical properties, and the 6-band version used as reference in this work. The numbers in the panel labels are the globally averaged differences.**

The results of this work and previous observation-based estimates are shown in Table 1. A large amount of the spread between estimates can be attributed to different global mean AOD, in particular those derived using MACC (Bellouin et al., 2013),
which had a significantly larger global-mean AOD at 0.18, compared to 0.12 for this work. To compare like with like, it is therefore useful to compare the radiative forcing efficiency, defined as the radiative forcing per unit optical depth. This is also shown in Table 1 where given in the cited literature, and suggest a much better agreement between studies. There is good agreement between the radiative efficiency estimates derived in this work and various literature estimates, including Bellouin et al. (2013). This confirms that differences with previous work are in great part due to differences in AOD and gives



confidence in our methodology and the representativity of the uncertainty estimates in the following sections. Figure 4 shows
the forcing efficiency at the TOA and surface derived in this work.

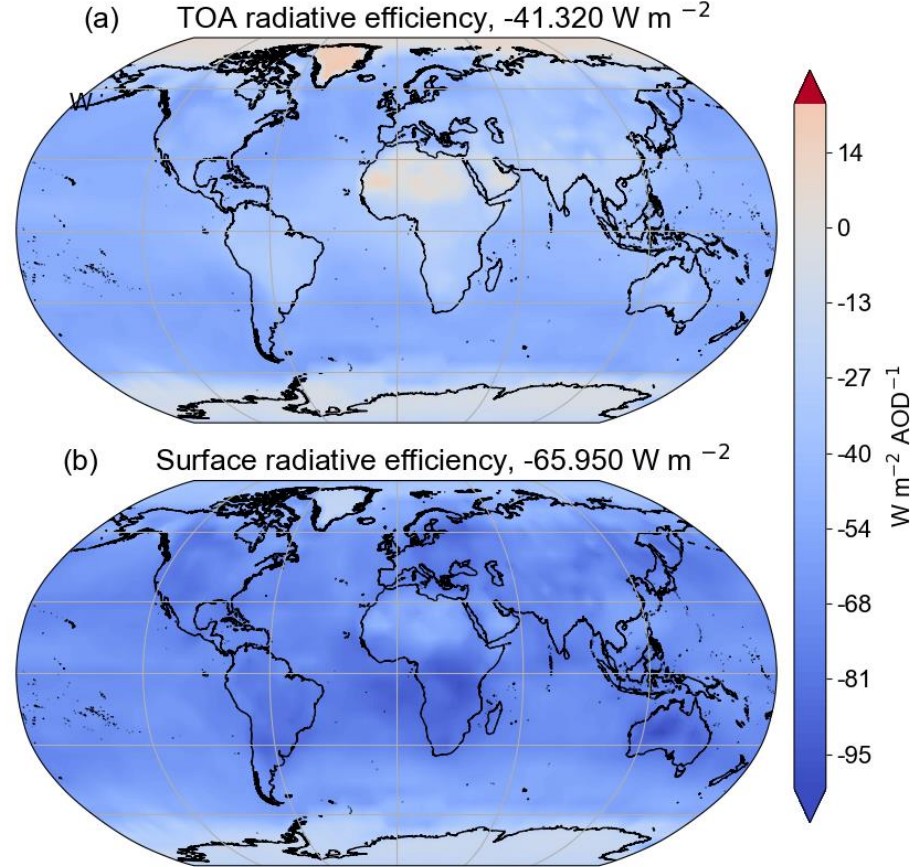

**Figure 4: TOA (panel a) and surface (panel b) annual-mean Direct Aerosol Radiative Forcing (DARF) efficiency, in W m$^{-2}$ per unit**
**AOD, as derived from MACv2 and SOCRATES. Global mean values are shown above each panel.**

**2.3. Uncertainties**

MACv2 does not contain direct information about uncertainties in aerosol optical properties. To obtain a realistic estimate of
the uncertainty in DARE and DARF due to those uncertainties, we assume a range of uncertainties in the columnar optical
properties, which are reflective of column-averaged upper and lower limits that are attainable from measurements, with a
balance struck between encapsulating the plausible range, and allowing for a large enough statistical sampling to obtain a
robust uncertainty estimate. The range and application of the uncertainties used is shown in Table 2. The uncertainties
described in Table 2 all refer to one sigma systematic (i.e., affecting every gridpoint equally) uncertainties. AOD uncertainties
are reflective of the spread in global satellite-derived uncertainty – the range chosen has a larger upper limit than the spread in



Figure 5 of Kinne et al. (2006). SSA and asymmetry factor uncertainties were selected to span the range of uncertainties from
Table 1 of Dubovik et al. (2002). The uncertainties in the optical properties are defined at 550 nm, and the fractional uncertainty
at 550 nm is then assumed for all other wavelengths. This will result in biases relative to an approach which has a more
sophisticated spectral treatment but is necessary for computational tractability. While the true uncertainty in DARE and DARF
is a function of many variables other than the aerosol optical properties, as discussed previously, here we only account for the
uncertainty attributable to the aerosol optical properties themselves. This choice matches our objective of quantifying the
reduction in uncertainty that could come from improved retrievals of AOD, SSA, and asymmetry parameter.

| Variable | Uncertainty range | Distribution |
|---|---|---|
| Aerosol Optical Depth (AOD) | 0.005-0.05 | Normal |
| Single-Scattering Albedo (SSA) | 0.01-0.04 | Lognormal in $(1 - \omega_0)$ |
| Asymmetry factor (g) | 0.01-0.08 | Normal |

**Table 2: Uncertainty ranges and statistical distribution shapes used for aerosol optical properties in the Monte Carlo experiments.**

For each given combination of the systematic uncertainties in Table 2, 500 perturbations to the optical properties are performed
in a Monte Carlo framework. Given the size of the parameter space to be sampled, the results given in this paper are made of
over 2.3 million gridded radiative transfer simulations, resulting in over 6 billion calls to the radiative transfer solver. Each
Monte-Carlo perturbation is made with vertically resolved optical properties at each point of the 5x5 lat-lon grid, which are
perturbed by the same relative amount. The same perturbation is applied to each of the calculations that make up the calculation
of the annual mean, to simulate a systematic uncertainty or bias that applies throughout the whole year. This ensures that any
resulting uncertainties in the TOA or surface DARE or DARF are not masked by compensating biases. The same perturbations
are applied to both present-day and pre-industrial aerosols. This ensures that the anthropogenic fraction remains constant,
meaning that any resulting uncertainty can be attributed solely to the aerosol optical properties. This methodology therefore
explicitly accounts for amplification or masking of the anthropogenic DARE by perturbations to the natural aerosol optical
properties, in contrast to some other estimates (e.g. Thorsen et al., 2021), in addition to any covariances between uncertainties
in different optical properties (e.g. an uncertainty in $\omega_0$ may have a different effect for different values of the uncertainty in *g*
and vice versa).

The uncertainty in the AOD in this work is taken to be representative of an uncertainty in the global mean AOD as measured
by satellites, because past observationally based estimates used AOD derived by satellites, rather than ground-based
photometers, for the sake of achieving global coverage. A draw is taken from a Gaussian distribution centred on the global
mean AOD with standard deviation $\sigma_{AOD}$ equal to the AOD uncertainty, and the ratio $\frac{AOD_{global}^{perturbed}}{AOD_{global}^{orig}}$ is used to perturb each



gridpoint by the same fractional amount for a given sample, so that the relative distribution of AOD remains constant. A gridpoint-wise (i.e. random) uncertainty is also applied depending on aerosol type, similarly to Bellouin et al. (2013), where:

$$\sigma_{AOD,random} = \begin{cases} 0.03 + 0.05 \cdot AOD \ (over \ ocean) \\ 0.03 + 0.15 \cdot AOD \ (over \ land) \end{cases}$$


The SSA uncertainty $\sigma_{\omega_0}$ is taken to be representative of the uncertainty in an inversion from a ground-based sun-photometer, e.g., from AERONET version 3 (Sinyuk et al., 2020) or GRASP (Dubovik et al., 2021). This is because SSA uncertainties remain better characterised in ground-based inversions than in the relatively recent satellite-based SSA products. SSA perturbations are applied separately to the coarse-mode, pre-industrial fine mode, and anthropogenic fine mode aerosols. These

perturbations are again spatially and temporally consistent.

An absolute change in SSA has more of an effect at large values (close to 1), since such a change will result in a larger proportional increase in the absorption coefficient. Additionally, SSA is constrained by the range $0 \leq \omega_o \leq 1$. Since typical values of $\omega_0$ are around 0.9 or above, a normal distribution in $\log(1 - \omega_0)$ is used, with perturbations transformed back into

$\omega_0$ and new absorption and scattering coefficients calculated. This approach is not without its limitations; at large values of $\omega_0$ such a lognormal distribution will result in significantly more extremal values than at lower SSA. Therefore, we assume that for regions where the SSA is large ($\omega_0 > 0.98$) for a given aerosol type, such as regions with high concentrations of sea salt and sulphate aerosols, there is no SSA uncertainty in that aerosol type and the SSA is not perturbed. This approach is similar to the hypothetical enhanced approach of Thorsen et al. (2021) and will result in reduced SSA uncertainties but ensures

that outliers drawn from such a distribution do not artificially increase the DARF uncertainty. The choice of probability distribution is somewhat subjective; this approach was chosen since it best retained the link between the input uncertainty and the width of the resulting probability distribution without the need for any tuned parameters.

A similar treatment is applied to the uncertainty in the asymmetry factor $g$. The uncertainty is assumed to be normally

distributed and systematic globally. Like in the SSA case, a different draw from the probability distribution is made for each aerosol type and sample, and applied globally, to ensure that the present-day natural and pre-industrial aerosols share the same perturbation.

Using this approach, it is possible to not only determine the relative importance of uncertainties in each of these three optical

properties, but also to determine the uncertainty in DARE and DARF obtainable by advances in measurements and retrievals of these optical properties. In addition, while in each of these scenarios the optical properties are perturbed within the same uncertainty limits globally, which is not necessarily realistic for measurements that may have different regional biases, the DARF uncertainty for each gridpoint is independent of its neighbours. Therefore, the output DARE and DARF uncertainties





in each single column can be combined, by mixing results from different sets of simulations to determine a more realistic
assessment of the global DARE/DARF uncertainties, as demonstrated in Section 4.3. A standalone software tool is provided
(see Section 6), which uses the simulations performed in this work to determine the resulting forcing uncertainty for a given
set of optical property uncertainties.

## 3.    Uncertainties in DARE and DARF

As with the reference case, for each perturbed set of aerosol parameters, the four radiative transfer calculations comprising the
seasonal averages are compared with either a no-aerosol case to compute DARE at the TOA and surface, or with a
corresponding perturbed pre-industrial case to obtain DARF, again at the TOA and surface. The resulting values of DARE and
DARF, either globally averaged or for a single column, are combined in a histogram with the standard deviation giving the $1\sigma$
uncertainty. Figure 5 shows an example of the global-annual mean TOA DARF for one set of input uncertainties, with Figure
6 showing the evolution of the standard deviation with respect to the number of samples. These Figures show a clear Gaussian
250    distribution (despite the distribution of $\sigma_{\omega_0}$ not being so) with little skewness and few outliers, with statistical stability to two
decimal-place precision at the TOA in the derived $\sigma_{DARF}$ after about 250 samples, a similar number to that found in Bellouin
et al. (2013).

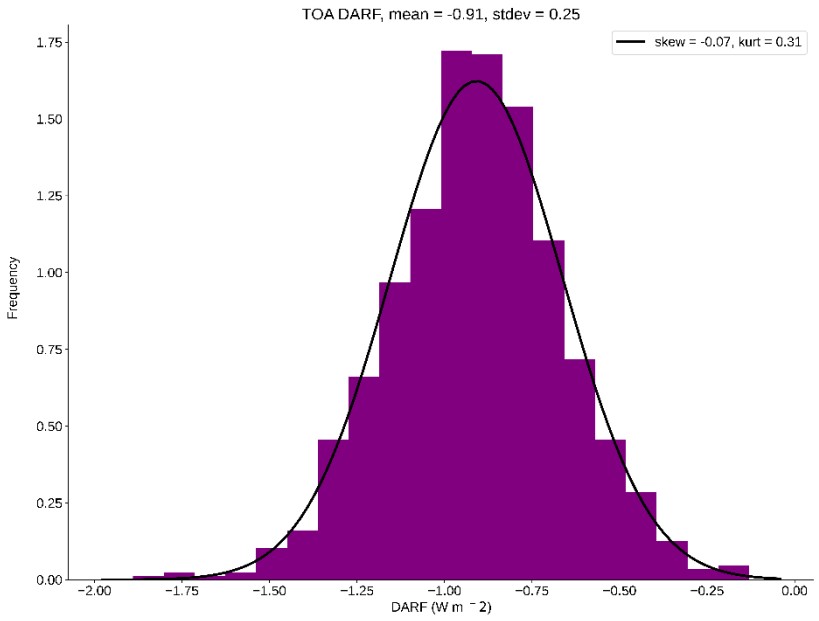




**Figure 5: Example histogram generated by 500 samples of global-annual mean Direct Aerosol Radiative Forcing (DARF) at the Top of the Atmosphere (TOA), in W m⁻², for a set of input uncertainties.**

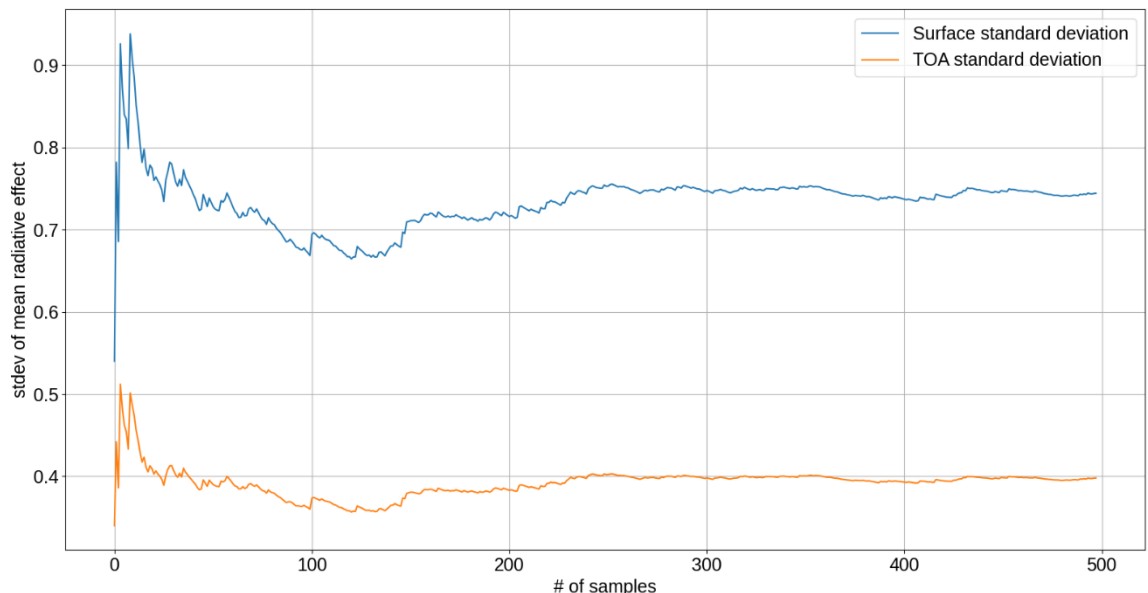

**Figure 6: Standard deviation of the histogram shown in Figure 5 as a function of the number of samples that make up the perturbed parameter ensemble for a given set of optical property uncertainties.**

Figure 7 shows several maps of TOA DARF uncertainty with respect to different choices of input uncertainties, in this case $\sigma_{AOD} = 0.02$ and $\sigma_g = 0.03$, with varying values of $\sigma_{\omega_0}$. The gridded uncertainties show the same spatial distribution as the forcing values shown in Figure 1, as would be expected, with $\sigma_{DARF}$ monotonic in $\sigma_{\omega_0}$.



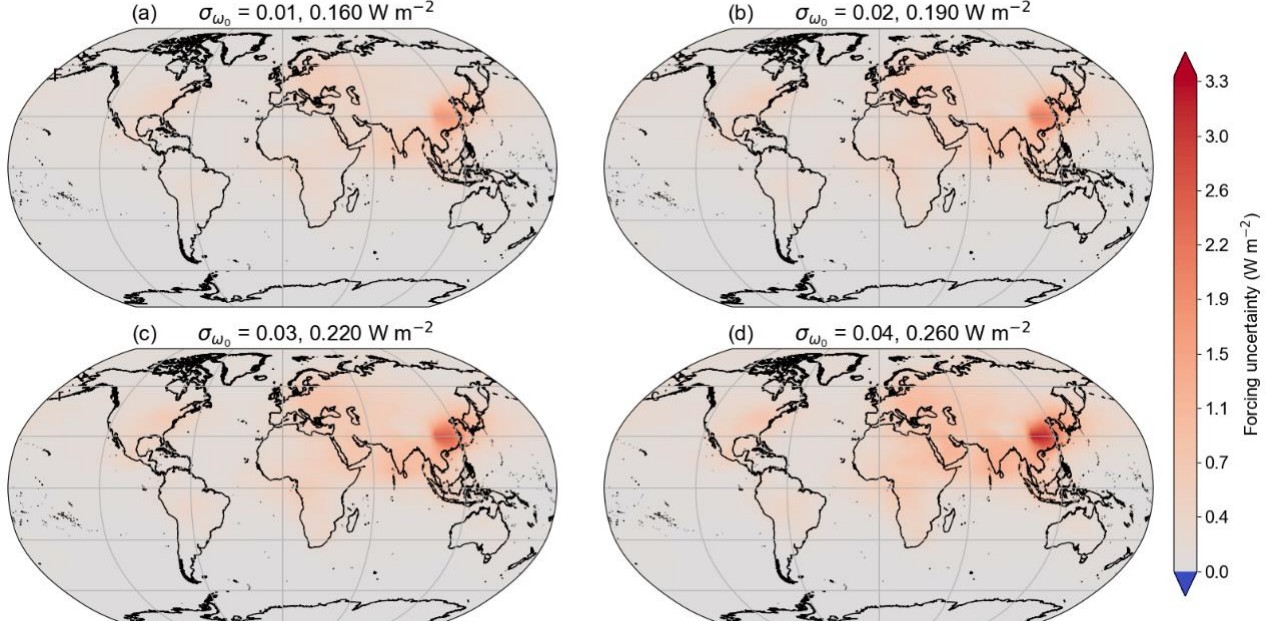

**Figure 7: Example maps of uncertainty in top-of-atmosphere Direct Aerosol Radiative Forcing (DARF), calculated as the standard deviation in W m$^{-2}$, generated using 500 Monte Carlo samples for four combinations of optical property uncertainties. Each panel has the same uncertainty in aerosol optical depth and asymmetry factor of $\sigma_{AOD} = 0.02$ and $\sigma_g = 0.03$, respectively, with increasing uncertainty in single-scattering albedo $\sigma_{\varpi_0}$ as indicated in the panel titles.**

## 3.1. Uncertainty ranges

Figures 8 and 9 show the global-annual mean TOA and surface DARF uncertainty derived using the uncertainty ranges in Table 1, with respect to $\sigma_{\omega_0}$ and $\sigma_{AOD}$. Each panel represents a step change in $\sigma_g$ from 0.01 to 0.08. There are several features of note in these Figures. At the TOA, the radiative forcing uncertainty appears to be roughly equivalently dependent on the uncertainties in AOD and SSA, with a smaller dependence on the uncertainty in asymmetry parameter. At the surface, the change in the DARF uncertainty is more clearly dominated by changes in the AOD uncertainty, as is the case when looking at DARE (see Supplementary Information). In each of these cases, there appears to be roughly equal weighting to increases in $\sigma_{\omega_0}$ and $\sigma_g$.

Several studies (Loeb and Su, 2010; Thorsen et al., 2021; Samset et al., 2018) demonstrated that $\sigma_{\omega_0}$ is a dominant source of uncertainty in TOA DARF; this may be due to their smaller global-mean AOD uncertainty than used in the ranges in Table 2, which are taken to be reflective of the spread in observed global-mean AOD from satellites rather than the uncertainty in AERONET retrievals as in Thorsen et al. (2021) and Dubovik et al. (2000). Figure 8 indicates that the uncertainty in TOA DARF is instead more *sensitive* to an uncertainty in the AOD, which is intuitive. The shape of the contours may be indicative of some covariance or non-linearity between the different optical property uncertainties in some cases, e.g., in panel (h) of



Figure 8. This may also be due to statistical anomalies due to insufficient sampling, however as demonstrated in Figures 5 and 6 this is unlikely to be a significant effect, since the derived $\sigma_{DARF}$ at both the TOA and surface appear to be stable after only 250 of the 500 samples run in each case.


The uncertainties in Figure 8 range from 0.08 - 0.47 W m$^{-2}$, i.e., ~9% - 52% relative uncertainty. This represents the upper and lower limits of what is feasibly attainable by hypothetical measuring systems capable of measuring globally with the uncertainties shown in Table 1. However, this is not necessarily representative of the real uncertainty, for two reasons. Firstly, this only accounts for uncertainties in aerosol optical properties. Host model uncertainties and uncertainties in variables not accounted for explicitly here (such as anthropogenic fraction) will still be present. In addition, Figures 8 and 9 assume that the absolute uncertainty is the same everywhere globally, which is not the case because uncertainties in AOD and SSA for example are anti-correlated (Dubovik et al., 2000), i.e., regions with low AOD (and correspondingly low absolute uncertainty in the AOD) will have large uncertainties in SSA. A more realistic assessment of uncertainty in DARE and DARF is given in Section 3.3.


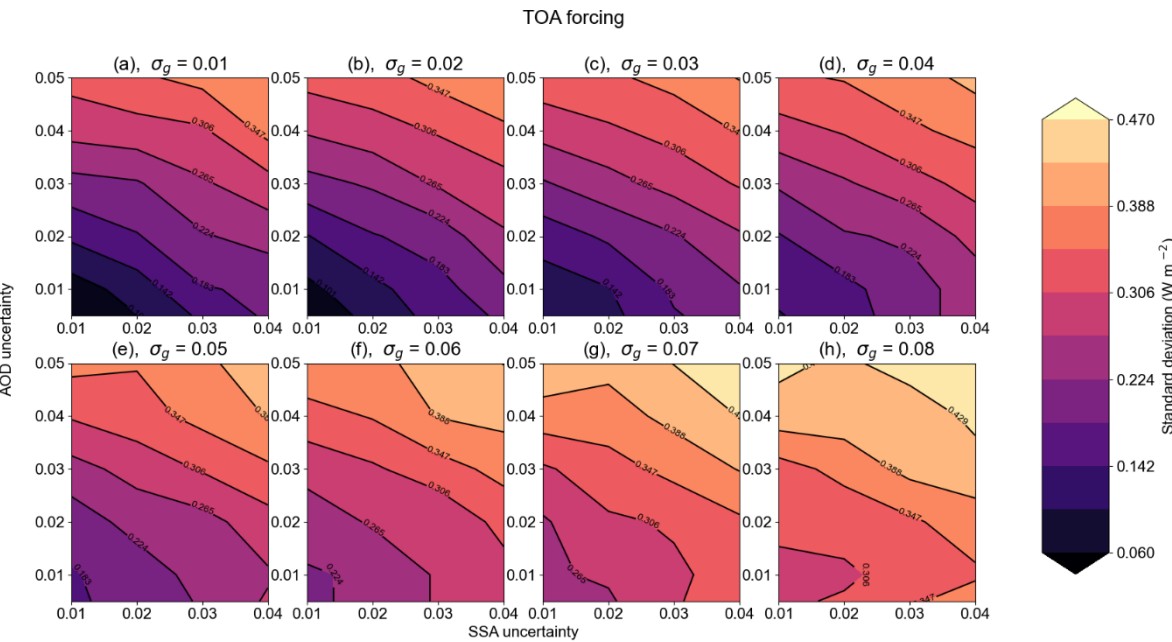

**Figure 8: Contour of top-of-atmosphere (TOA) Direct Aerosol Radiative Forcing uncertainty, in W m$^{-2}$, with respect to SSA (x-axis) and AOD (y-axis) uncertainty. Each panel represents a change in asymmetry factor uncertainty of 0.01, within the 0.01 to 0.08 range.**






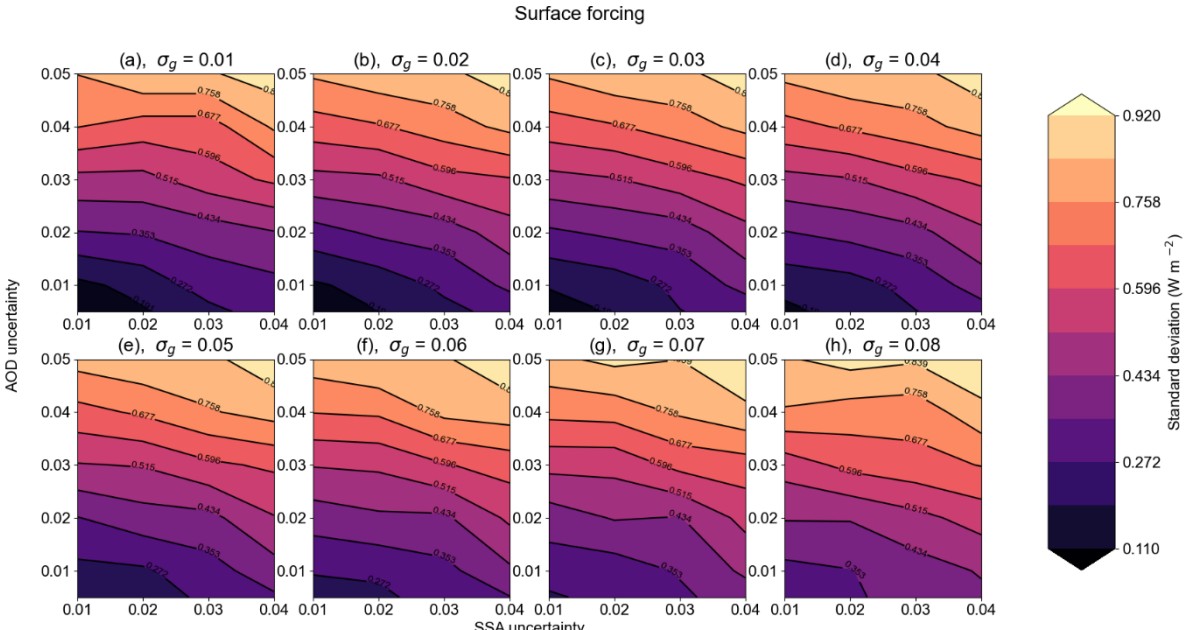

**Figure 9: As Figure 8, but for surface Direct Aerosol Radiative Forcing.**

**3.2. Attribution of DARE and DARF uncertainty to uncertainties in optical properties**

The results in Section 3.1 are idealised, and provide a broad range in which $\sigma_{DARF}$ and $\sigma_{DARE}$ are likely to sit as a function of only uncertainties in the optical properties. It is instructive to determine the sensitivity of $\sigma_{DARF}$ to each of the input uncertainties in turn. Assuming linearity and no covariance, the sensitivity coefficient $c_x$ is simply given as

$$c_x = \frac{\partial \sigma_{DARF}}{\partial \sigma_x}$$

where $x$ is either AOD, $\omega_0$ or $g$. It is worth bearing in mind that this is done for heuristic purposes – as shown on Figures 8 and 9 even on the global mean scale significant nonlinearities/covariances exist, which may be stronger locally. Nevertheless, this gives an indication of which uncertainties are strongest in which regions, and therefore where the most value can be obtained by increasing precision in a given variable.

The results for TOA and surface DARE are shown in Figure 10, and Figure 11 shows the corresponding results for TOA and surface DARF. These two Figures show a number of interesting features. Figure 10 shows that the DARE for both the TOA and surface is much more sensitive to $\sigma_{AOD}$ than $\sigma_{\omega_0}$ and $\sigma_g$. However, over desert regions the uncertainty in SSA dominates at the TOA (Figure 10, panel a). This is due to the combination of a more strongly absorbing aerosol over a highly reflective surface. It may also be due to limitations in the modelling framework, as coarse dust aerosols tend to have a lower Ångström





exponent, i.e. have a larger AOD at longer wavelengths, and the fractional uncertainty is assumed equal at all wavelengths. It may also be due to covariant effects (e.g. increased scattering from SSA offset by increased backscatter from asymmetry), which are not accounted for in this analysis.

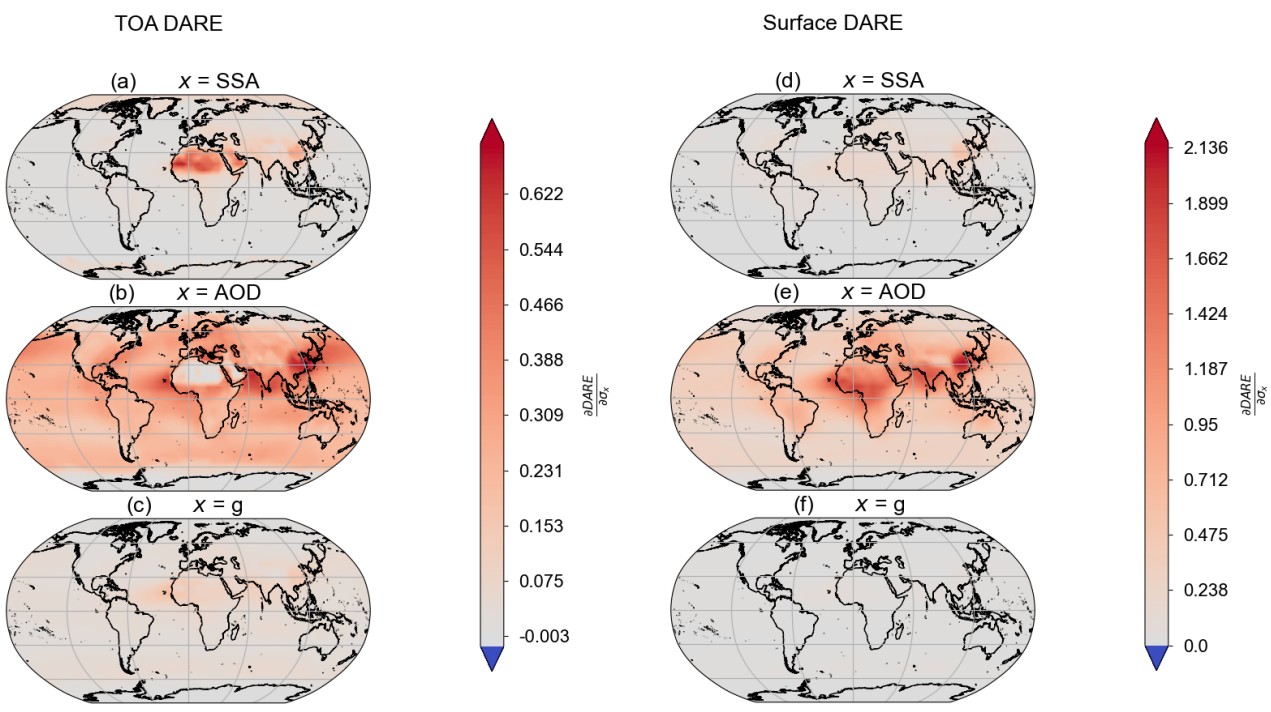

**Figure 10: Sensitivity of Direct Aerosol Radiative Effect (DARE) at top-of-atmosphere (TOA) (left) and surface (right) to uncertainties in aerosol optical depth (AOD), single-scattering albedo ($\omega_0$) and asymmetry parameter ($g$). Units are W m$^{-2}$ per unit optical property uncertainty.**

Figure 11 panels (a) and (b) show that the sensitivities of DARF uncertainty to AOD and SSA are very similar for most regions,

particularly the strong anthropogenic forcing over East Asia, but with stronger effects from SSA over desert regions and stronger sensitivity to AOD over regions with significant anthropogenic aerosol, such as Southern Africa and North America. This reflects the generally even shape of the contours on Figure 8. Panel (c) of Figure 11 shows that the asymmetry factor uncertainty is also important, but less so than SSA and AOD. At the surface, the uncertainty in surface DARF is almost entirely insensitive to $\sigma_g$, aside from the region of strong anthropogenic emissions over East Asia, and much more sensitive to $\sigma_{AOD}$

than $\sigma_{\omega_0}$ as shown in Figure 9. In both cases, one expects a first-order cancellation of the radiative effects of the present-day





natural and pre-industrial aerosols, which are the same in our framework, so their uncertainties do not matter much for DARF, in constrast to DARE.

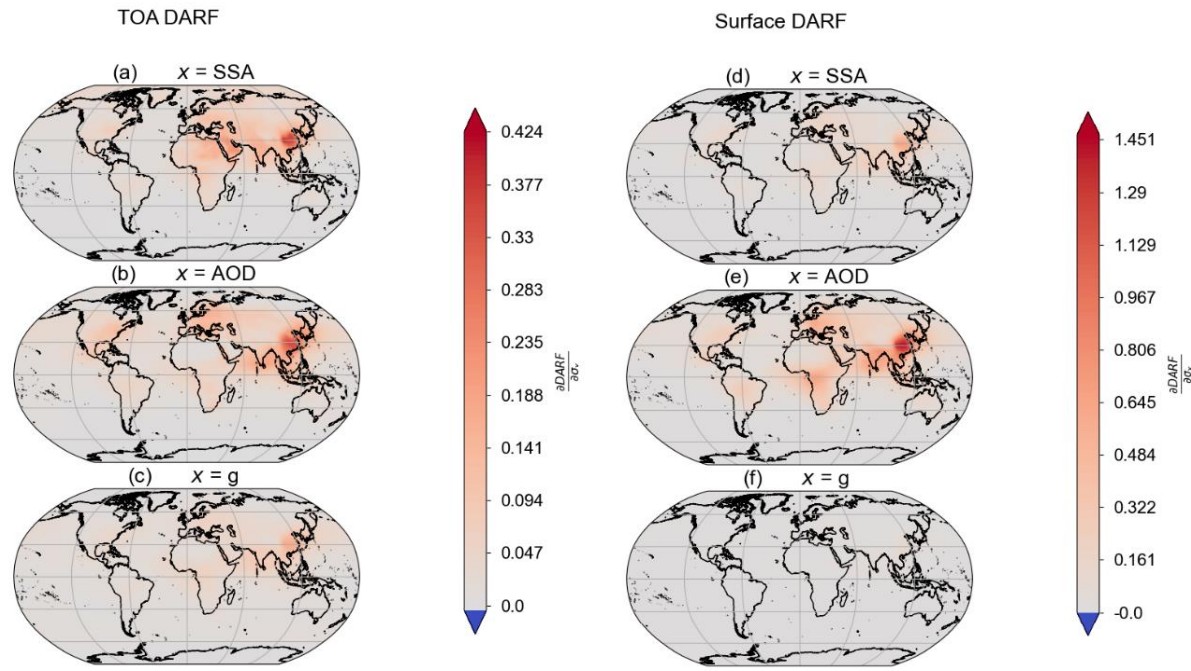

**Figure 11: Same as Figure 10, but for Direct Aerosol Radiative Forcing (DARF).**

Figure 12 shows the largest contributor to the uncertainty among AOD, $\omega_0$ and $g$, for TOA and surface DARF and DARE. At the surface, the main contributor to uncertainty is AOD almost everywhere on the globe, for both DARE and DARF. At the TOA a more complex picture develops. For DARE, the uncertainty in SSA is dominant in regions with high surface albedo

such as deserts and at the poles, with AOD being most important elsewhere. For DARF, the main contributor varies regionally, but AOD generally dominates, except again over bright surfaces, and over remote, low-AOD regions where scattering dominates due to high SSA sea-salt aerosol, where $g$ dominates. Previous studies (Loeb and Su, 2010; Samset et al. (2018); Thorsen et al., 2021) found that SSA uncertainties were more dominant, but used smaller AOD uncertainties, based on the abilities of ground-based sun-photometers rather than those of satellite retrievals.






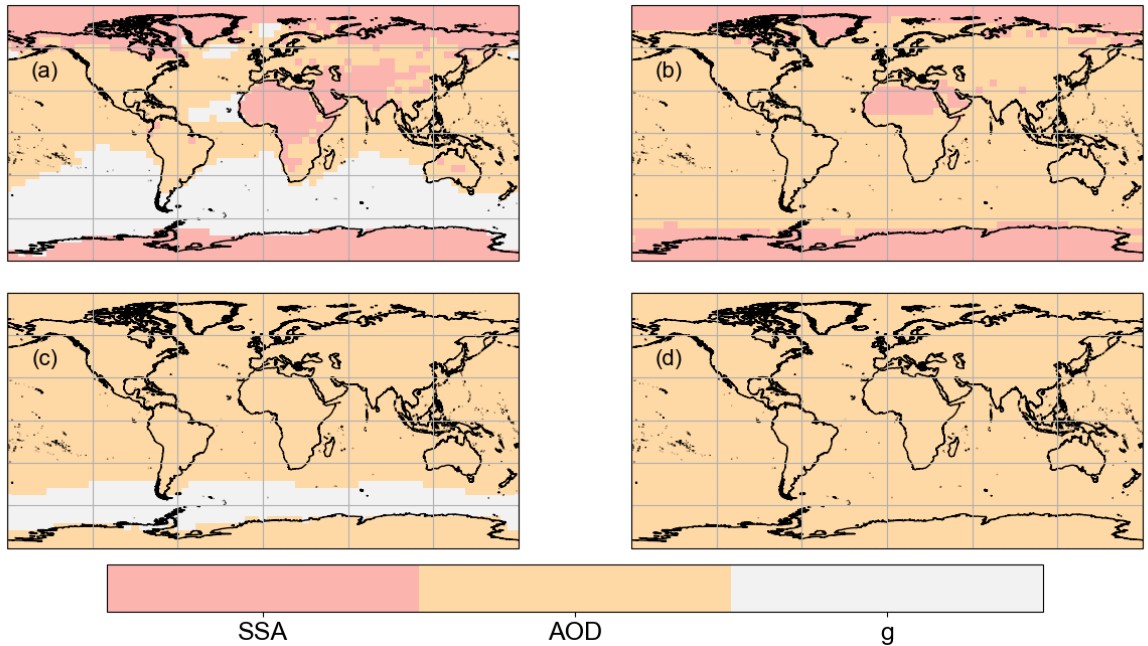

**Figure 12: The largest contributor to the uncertainty in each of the three single scattering properties for four different cases: top-of-atmosphere (TOA) Direct Aerosol Radiative Forcing (DARF, panel (a)), TOA Direct Aerosol Radiative Effect (DARE, panel (b)), surface DARF (panel (c)) and surface DARE (panel (d)).**

## 3.3. Regionally based estimates

The analysis has so far assumed that uncertainties are globally uniform. But the lookup tables derived in Section 3.1 can also be used to obtain regionally based estimates of the DARE and DARF uncertainty using regionally varying estimates of $\sigma_{AOD}$, $\sigma_{\omega_0}$ and $\sigma_g$. As stated in Section 3.1, Figures 8 and 9 are global-mean representations of thousands of similar such plots corresponding to each gridpoint. We can therefore get a more realistic estimate of the uncertainty in DARE or DARF by selecting the point in each of these individual contour maps that correspond to a user-defined uncertainty in the aerosol optical properties in each gridbox from a lookup table, and recombining them to generate a new global-annual mean. By attributing each gridpoint a specific $\sigma_{AOD}$, $\sigma_{\omega_0}$ and $\sigma_g$, it is possible to obtain a more realistic estimate of $\sigma_{DARE}$ and $\sigma_{DARF}$. This section outlines an example of this approach, using optical property uncertainties similar to Bellouin et al. (2013, hence B13), which are based on AERONET v1 uncertainties (Dubovik et al. (2002)). The software and data used to obtain this estimate, and to generate such estimates for other sets of input uncertainties are available for download (see the Data Availability section). Since B13 only define uncertainties for anthropogenic aerosols, the uncertainty in $\sigma_{\omega_0}$ and $\sigma_g$ in each gridpoint is scaled via the anthropogenic AOD fraction at 550 nm for consistency. $\sigma_{AOD}$ is fixed to 0.03 everywhere as in B13, i.e., we similarly assume that all the uncertainty is due to anthropogenic aerosol for this case.



B13 and this work differ significantly in their methodological frameworks, despite both having DARF uncertainties derived
via Monte Carlo sampling of the input uncertainties. Particularly relevant to this comparison is that B13 include several
uncertainties not factored in here, such as uncertainties in anthropogenic fraction (Table 1 of B13). Additionally, B13 and this
work use significantly different methods to determine aerosol type, with B13 using a bespoke algorithm on MACC reanalysis
data compared to the prescribed aerosol optical depth used in MACv2. Nevertheless, the use of similar optical properties allows

for a direct comparison.

The DARE and DARF uncertainties obtained via this approach are shown on Figure 13. Global averages can be compared
with those given in Table 2 of B13. At the TOA, the uncertainty in this work ($\pm$ 0.22 W m$^{-2}$) is significantly smaller for DARF
("anthropogenic DRE" in B13, $\pm$ 0.5 W m$^{-2}$), even when scaling the values and their associated uncertainties by the global-

mean AOD (i.e., simply scaling the B13 values by a factor of 0.66, giving $\pm$ 0.33 W m$^{-2}$). Similarly, the surface DARF
uncertainty is significantly lower in this work ($\pm$ 0.49 W m$^{-2}$) than B13 ($\pm$1.1 W m$^{-2}$, $\pm$0.73 W m$^{-2}$ when scaled). Taking both
sets of uncertainties at face value, this would suggest that the optical properties account for around 40-60% of the total
uncertainty in the aerosol radiative forcing at both the TOA and surface, with the remainder being the result of other
uncertainties (i.e., anthropogenic fraction) considered in B13. For DARE, the results of this work and B13 are much more

similar ($\pm$ 1.09 vs. $\pm$ 1.3/$\pm$ 0.86 W m$^{-2}$ at the TOA, $\pm$ 1.97 vs. $\pm$1.9/1.26 W m$^{-2}$ at the surface), likely due to the anthropogenic
fraction being a second-order contributor to the uncertainty in DARE, as would be expected.

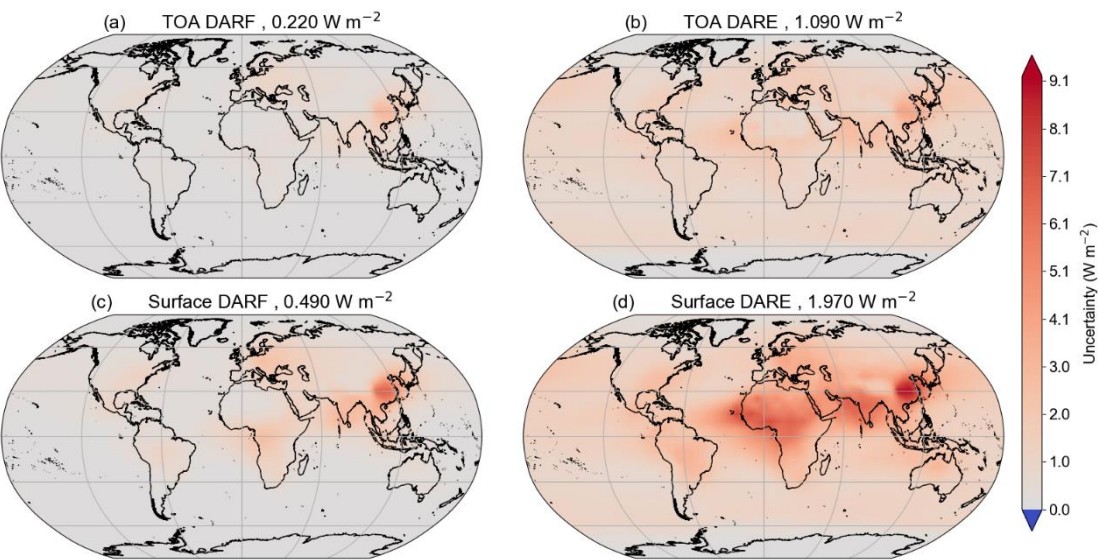

**Figure 13: Top-of-atmosphere (TOA, top) and surface (bottom) Direct Aerosol Radiative Forcing (DARF, left) and Effect (DARE, right), in W m$^{-2}$, as derived using regional optical property uncertainties from AERONET v1. Global average values are shown in**

**the panel titles.**



It is also possible to use the sensitivity coefficients derived in Section 3.2 to obtain a similar estimate of the uncertainty to that obtained using the lookup table approach (Section 3.1). This allows us to look at the effect of nonlinearities for a specific example. Figure 14 shows this method applied to the TOA forcing uncertainty derived using the input uncertainties described

earlier in Section 3.3. The two approaches differ by 0.04 W m$^{-2}$, indicating that nonlinearities account for roughly 20% of the total uncertainty.

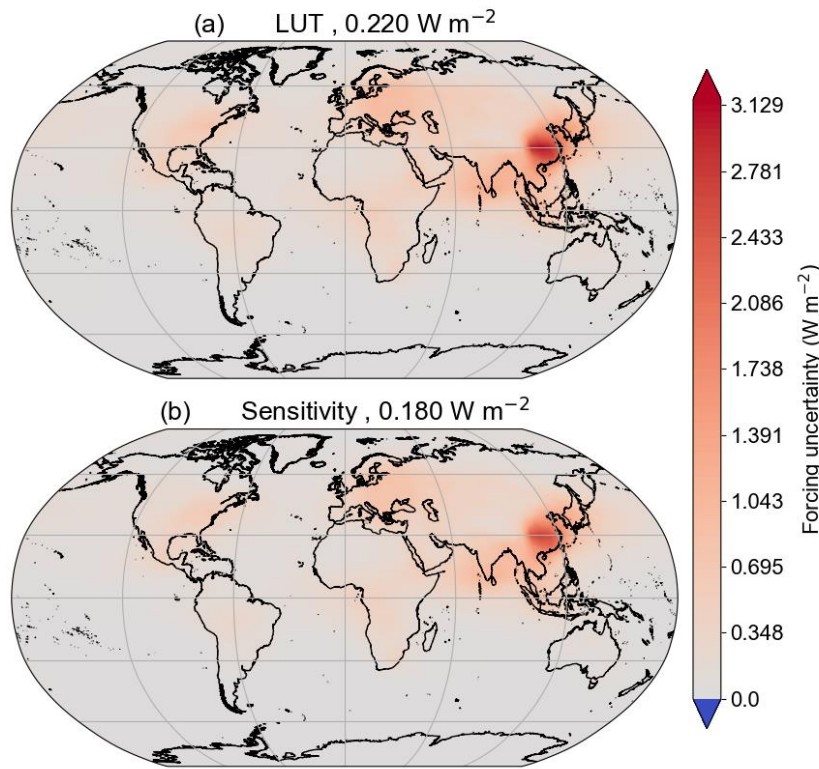

**Figure 14: Top-of-atmosphere Direct Aerosol Radiative Forcing, in W m$^{-2}$, derived using (top panel) the lookup table approach (Section 3.1) against that derived using (bottom panel) the sensitivity coefficients derived in Section 3.2 for the AERONET v1**
**example described in Section 3.3. Global mean values are shown in the panel headings.**



## 4. Scaling to all-sky conditions

The central estimate and uncertainty limits obtained in Sections 2 and 3 are applicable only to clear (cloud-free) skies. It is however possible to scale these uncertainties scale to all (clear and cloudy) skies at the TOA by using a cloud fraction distribution to scale the gridbox-level DARE or DARF for each of the four seasonal-mean calculations that make up an estimate of the global annual mean. To do this, we use monthly mean cloud fraction and cloud optical depth, $\tau_{cloud}$, from the International Satellite Cloud Climatology Project (ISCCP) H Series (Rossow et al., 2016) over the period 1983-2017. They are combined to obtain a present-day seasonal average cloud fraction and optical thickness, interpolating to the 5°x5° latitude-longitude grid used in the radiative transfer simulations. We assume that the DARE is entirely masked by clouds when optically thick ($\tau_{cloud} > 1$) clouds are present, but that optically thin clouds ($\tau_{cloud} < 1$) do not mask the DARE and leave it unchanged. The TOA DARE and DARF (and the calculation of the associated uncertainties) are then scaled by the cloud fraction in grid cells where the cloud is optically thick. These assumptions are not entirely correct, as it is known that above-cloud aerosol-radiation interactions occur from biomass-burning aerosols in cloudy regions. However, the corresponding cloudy-sky DARE is likely small globally, with Myhre et al. (2020) estimating a global average of only 0.01±0.1 W m$^{-2}$. Additionally, optically thin clouds will serve to mask some of the aerosol effect, further constraining both our central estimate and the uncertainty. We do not consider any uncertainties due to the clouds themselves here; this analysis is purely to scale the global-mean uncertainty in the aerosol optical properties. Other estimates that use a more sophisticated cloud representation have an increased uncertainty in all-sky conditions to reflect uncertainties in cloud properties but also the increased uncertainty associated with aerosol retrievals in cloudy conditions (Kacenelenbogen et al., 2019). It would be possible to account for these effects using our framework, by considering that the optical property uncertainties in cloudy regions are larger than those in clear-sky regions.

The results of the scaling described above are shown in Table 3, alongside the latest estimate from the Intergovernmental Panel on Climate Change AR6 report (Forster et al., 2021) and other recent studies that provide all-sky DARE and DARF estimates. The central estimate obtained with our approach is in very good agreement with the other estimates, with almost identical results to those of Kinne (2019b) and sitting in the range of plausible values implied by the various studies. However, our uncertainties are significantly lower than all other cases, since we only account for uncertainty in clear-sky aerosol optical properties. Taken together, Table 3 suggests that aerosol optical property uncertainty accounts for a third to half of total uncertainty.

| Study | DARE (W m$^{-2}$) | DARF (W m$^{-2}$) |
|---|---|---|
| This work (B13 uncertainties) | −1.87 ± 0.45 | −0.35 ± 0.09 |
| Kinne (2019b) | −1.8 | −0.35 (−0.2 < x < −0.45) |





| Bellouin et al. (2013) | n/a | $-0.7 \pm 0.2$ |
| Thorsen et al. ( 2021) | $-1.46 \pm 0.47$ (0.29) | $-0.26 \pm 0.31$ (0.19) |
| Matus et al. (2019) | $-2.40 \pm 0.6$ | $-0.50 \pm 0.3$ |
| IPCC AR6 | n/a | $-0.25 \pm 0.2$ |

**Table 3: Top-of-atmosphere, all-sky Direct Aerosol Radiative Effect (DARE) and Forcing (DARF), in W m$^{-2}$, for this work and selected comparable previous studies, along with their uncertainties where available. Uncertainty estimates for this work are obtained using the optical property uncertainties from Bellouin et al. (2013). The numbers in brackets for Thorsen et al. (2021) are the uncertainties in their "enhanced" methodology (see Section 1). Uncertainties from Kinne (2019b) are asymmetric, with −0.35 being the central value.**

## 5. Conclusion

Despite several decades of research uncertainties in DARE and DARF remain large (Forster et al., 2021). Based on plausible measurement uncertainties in AOD, SSA and asymmetry parameter, we quantified shortwave clear-sky, TOA and surface, DARE and DARF uncertainties. We used a new Monte Carlo framework, available for download, applied to over two million radiative transfer simulations using the radiative transfer code SOCRATES. We first assume uniform uncertainties globally, then use regionally varying uncertainties. Results are summarised in Table 1. When using globally uniform uncertainties, aerosol optical property uncertainties represent between 5 and 42% of DARE and 9 and 52% of DARF uncertainty at the TOA. At the TOA, AOD uncertainty is the main contributor to overall uncertainty, except over bright surfaces where SSA uncertainty contributes most. When using regionally varying uncertainties, aerosol optical property uncertainties represent 24% of TOA DARE and DARF. Clear-sky results are then scaled to all-sky conditions by scaling by ISCCP cloud fraction and assuming that cloud with an optical depth larger than 1 totally mask the DARE. Under these assumptions, aerosol optical property uncertainty contributes to about 25% uncertainty in TOA, all-sky DARE and DARF. Comparing our uncertainties, which only include the contribution of AOD, SSA, and asymmetry parameter uncertainties, to uncertainties obtained in previous studies, which also considered uncertainties in non-aerosol variables, suggests that the aerosol optical property uncertainty accounts for a third to a half of total uncertainty. This result suggests that reducing aerosol retrieval uncertainties, both for ground-based sun-photometers and satellite instruments, needs to be done in combination with reductions in non-aerosol uncertainties, such as surface and cloud properties.

Figure 15 shows estimates of the TOA DARF and its uncertainties, in clear and all-sky conditions, for the studies presented in Tables 1 and 3. The estimates for the present work correspond to the lower and upper limits of our globally uniform aerosol optical property uncertainties, and the regionally varying uncertainties from Section 3.3. Using the ranges of uncertainty tested



in this work, the smallest reasonable uncertainties in the optical properties ($\sigma_{AOD} = 0.005$, $\sigma_{\omega 0} = 0.01$ and $\sigma_g = 0.01$) result in a

clear sky TOA DARF uncertainty of $\pm 0.08$ Wm$^{-2}$. This value is significantly smaller than the range of uncertainties estimated from existing studies shown in Figure 15 and Table 1, which spans $\pm 0.16$ to $\pm 0.50$ Wm$^{-2}$. This again suggests that reducing aerosol optical property uncertainty would only reduce overall DARF uncertainty by up to a half. Using our regionally varying aerosol optical property uncertainties based on AERONET v1, the TOA DARF uncertainty of $\pm 0.22$ Wm$^{-2}$ is 2.75 times larger than our minimum value and is broadly consistent with the values derived in the radiative kernel study of Thorsen et al. (2021),

which range from 0.22 to 0.31 W m$^{-2}$. A benefit of our framework is that it allows for a quick assessment of the impacts of reduced uncertainties in AOD, SSA, and asymmetry parameter in specific regions, which can help inform which regions and variables go furthest to reduce the global-mean uncertainty. For example, if we divide by two the regionally varying uncertainties used in Section 3.3, the uncertainty in TOA DARF would be reduced from 0.22 (25% of DARF) to 0.12 W m$^{-2}$ (13% of DARF), a factor of just under 2.


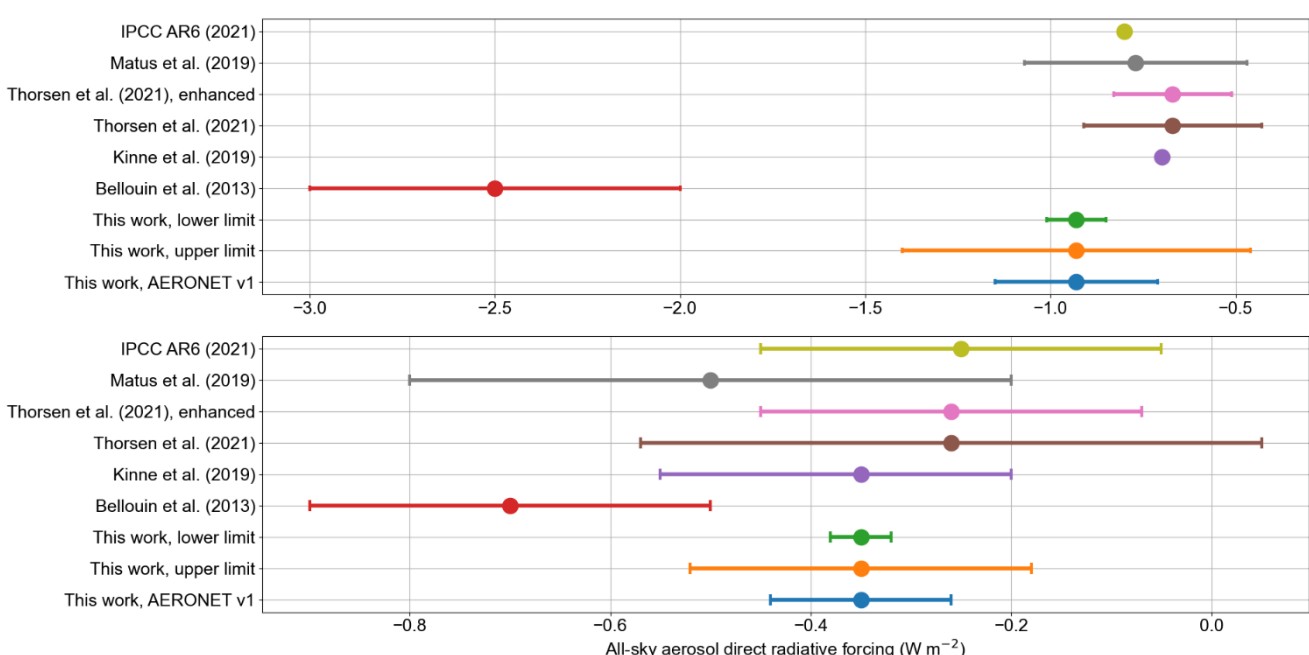

**Figure 15: Clear-sky (upper) and all-sky (lower) TOA forcing estimates, in W m$^{-2}$, and their 1-sigma uncertainties (where available) from this work compared with previous studies.**

There are several caveats that need to be considered when using the results from this work. While we can sample the parameter

space of the aerosol optical properties to a reasonable degree, there are some components to the uncertainty which we do not represent. Most notably, we assume that the aerosol property uncertainty is proportionally equal at all wavelengths. This is primarily for reasons of computational tractability, as applying different perturbations to different wavelengths would significantly increase the number of radiative transfer calculations required. A future experiment might include some



uncertainty in the extinction and/or absorption Angstrom exponents. Additionally, we do not represent the longwave component of DARE or DARF here. This is most relevant for coarse mode aerosols such as mineral dust and sea salt, and therefore for estimating DARE. Due to the net positive longwave DARE of coarse aerosols at the TOA (e.g. Ryder 2021), the total (i.e. shortwave plus longwave) DARE would be less negative than values given here overall, when incorporating longwave effects. DARF, in contrast, is dominantly determined by changes in fine to accumulation mode aerosols (making up the anthropogenic component of aerosol species), which mostly impact the shortwave spectrum, as represented here. Finally,

we solely focus on the *direct* radiative effect and forcing, neglecting the effects of aerosol-cloud interactions, all of which need to be captured to fully represent the effects of aerosols on climate. Nevertheless, the relative uncertainties in the DARF are as large as those due to aerosol-cloud interactions (Forster et al., 2021), rendering it important to understand the contributors to these uncertainties.

Recent progress in constraining optical properties, such as from AERONET v3 (Sinyuk et al., 2020) and GRASP (Herrera et al., 2021) could further reduce the TOA DARF uncertainty, although as noted above non-aerosol uncertainties contribute substantially to total uncertainty. Our results provide a framework within which future new measurement uncertainties can be evaluated globally to estimate their impact on global DARE and DARF uncertainty, such as those from upcoming missions such as EarthCare (Wehr et al., 2006) and Plankton, Aerosol, Cloud, Ocean Ecosystem (PACE; Werdell et al., 2019).

**6.  Code/Data Availability**

The sensitivity data and our Monte-Carlo uncertainty framework tool are available in the MAPP project Zenodo repository at https://doi.org/10.5281/zenodo.7958296.

**7.  Author Contributions**

CR and NB conceived the study. JE constructed the experiments and code and performed the analysis with input from CR and
NB. JE wrote the manuscript, with comments and contributions from CR and NB.

**8.  Competing Interests**

The authors declare that they have no conflict of interest.



## 9. Acknowledgements

The authors acknowledge support from the European Metrology Program for Innovation and Research (EMPIR) within the
joint research project EMPIR 19ENV04 MAPP "Metrology for aerosol optical properties". The EMPIR is jointly funded by
the EMPIR participating countries within EURAMET and the European Union.



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
