# Peer review of "Sensitivity of global direct aerosol shortwave radiative forcing to uncertainties in aerosol optical properties"

_EGUsphere, 2023_

## Referee Comment (RC2)

**Sensitivity of global direct aerosol radiative forcing to uncertainties in aerosol optical properties** by J. Elsay et al.

**Strengths**
statistical approach (to capture impact uncertainty through multiple simulations)
simplicity (differences from dual/triple calls)
link to observational capabilities (sat AOD retrievals, absorption from Anet inversions)

**Weaknesses**
implications to climate prediction uncertainty are overdrawn (without indirect effects)
all-sky impacts (clear-sky to all-sky) are based on a simple approximation
pre-industrial aerosol properties are not (but should be) considered in DARF
DARE uncertainties do not include longwave effects (thus likely could be too large)

The paper applies aerosol radiative properties of the MACv2 aerosol climatology as a base-line for uncertainties of aerosol radiative effects and aerosol forcing – but only for solar impacts under cloud-free conditions (at TOA and surface) although with an approximation to estimate all-sky impacts. This study is interesting, but there are also aspects that need to be clarified and possibly corrected during the review process.

While MACv2 aerosol properties are applied to define the solar spectral aerosol properties (AOD, SSA, ASY) and the AOD vertical distribution, questions remain as to why the calculated results are so different from 'official' MACv2 simulations. For clear-sky solar (all) aerosol radiative at the surface the global average -8.6 W/m2 cooling is larger than -7.4 W/m2 in MACv2 (and -7.2 W/m2 in MACv3). This suggests that applied solar AOD values of the implementation in this study are larger. This then could also explain – with a similar absorption potential - why TOA impacts at -4.5W/m2 are more negative (than -3.5W/m2 in MACv2 and -3.7W/m2 in MACv3). It would be very useful to compare global 'input' AOD maps (especially for solar spectral AOD data) to identify the cause for these differences - under the premise that an identical input to aerosol properties was used. Suggested contributions of (environmental) non-aerosol properties are weak, because DARF and DART estimates are based on differences from two simulations, where these properties do not vary (just largely cancel in their impacts).

The estimated aerosol property uncertainty impact on DARE ignored longwave effects, which possibly could reduce uncertainties. For climate relevant DARF (at TOA) the solar impact is sufficient, but other important elements (the pre-industrial uncertainty and more important indirect effect through clouds) are not included, so that climate uncertainty relevance (for me) is more than limited.

Finally the applied observational uncertainties should be reviewed. For AOD I would include an uncertainty that increase also with AOD and for SSA I would ties uncertainty directly to AOD … possibly by using AAOD (AOD*[1-SSA]) uncertainty instead.

Overall, it is an interesting sensitivity study.

**Details**

(while reading the paper)

Line 33      Aside from TOA and surface impacts also atmospheric impacts and uncertainty could be addressed. Differences between TOA and surface indicate atmospheric impacts and quantify (for solar aerosol radiative effects) the solar heating in the atmosphere.

Line 70      the definition of anthropogenic aerosol requires a pre-industrial state, which cannot be measured and it largely drawn from model simulations (which seem reasonable but involve many assumptions). Given the uncertainty to wildfires at pre-industrial conditions, the anthropogenic AOD is relatively large not only for direct but also for indirect effects.

Line 72/95    I attach a netcdf file with MACv3 AOD/SSA/ASY data at 550nm and I like the authors to check if their implementation reveals similar (especially) AOD values (because I just do not understand why the radiative effects are so more negative – e.g. the DARE values to the net-fluxes at the surface – as the implementation, as described sounds reasonable. If anymore comparison data (e.g. surface albedo, solar insolation) should be compared, please let me know.

Line 129      the explanation attempt via environmental differences seems unlikely as difference in the set-up will cancel when subtracting two simulations.

Line 150      As expected the solar spectral resolution is not an issue, but thanks for checking.

Line 165      The forcing efficiencies agree much better than the forcing … so AOD field difference seem the most likely explanation (I suggest to compare your AOD data to those of an .nc MACv2 or MACv3 data files, which is will send the authors on request).

Line 183      I would associate the AOD not just with a fixed deviation, but also as a function of AOD (as in many satellite retrieval evaluations). Similarly I also would associate the SSA uncertainty with AOD, as suggested by Dubovik. In that sense I would work with AAOD uncertainties rather than SSA uncertainties (and yes satellite data have little skill and mainly reflect a priori assumption). I do no care so much about ASY uncertainties Another way for uncertainties would be to work with local ranges of multi-annual monthly (or in your case seasonal) data at least for AOD, which are provided from ICAP satellite data multi-model assimilations.

Line 272/274 I agree with that assessment

Line 290    For DARF I would distinguish between anthropogenic fraction and other 'host model uncertainties', because the host uncertainties largely cancel via differences from two simulations, whereas 'anthropogenic' does not.

Page 294    … and for that reason use (for absorption uncertainty) AAOD uncertainties rather than SSA uncertainties. And SSA uncertainties to DARF and DART are less meaningful at low AOD.

Page 315    the dust SSA uncertainty is mainly associated with uncertainties to dust size. But also here I see as AOD association as the larger the dust AOD, the larger the dust size and with that the lower the mid-visible SSA.

Page 385    I agree that anthropogenic uncertainties for DARF should be large. Why was that not included this in your simulations? Pre-industrial AOD and SSA uncertainties are MUCH larger than current satellite or Aeronet uncertainties.

Page 410    MACv3 also uses ISCCP but distinguishes between high, mid and low altitude cloud cover (impacts on forcing and forcing efficiency can be taken from the images). This approach here is simpler but ratios could be compared to MACv3.

Page 430    MACv3 is more absorbing in the fine-mode so that DARF is now reduced from -0.35 to -0.23 W/m2 (just as info). Surface DARF is with -1.9 W/m2 similar.

Page 442    Since only solar radiative transfer is addressed here I would focus on DARF (and not so much on DARE).

Page 471    all-sky DARF in MACv3 compared to MACv2 is now shifted to less negative values (based on the same arguments). Still these uncertainties are small compared to uncertainties from indirect effect though water clouds (cover, optical depth, droplet radius). In the conclusions this should be put in perspective … and possibly a follow-up study (to include indirect impacts) on DARF?

**MACv3**

[Figure]

The annual average maps of the radiative effects with MACv3 (similar to MACv2) at TOA (rows 1/3) and surface (rows 2/4), for clear-sky (rows 1/2) and for all-sky (rows 3/4) for solar and IR combined (col1), for solar only (col2) and for anthropogenic solar (col3)

[Figure]

same as above … now for the forcing efficiencies with MACv3

**MACv2**

[Figure]

The annual average maps of the radiative effects with MACv2 at TOA (rows 1/3) and surface (rows 2/4), for clear-sky (rows 1/2) and for all-sky (rows 3/4) for solar and IR combined (col1), for solar only (col2) and for anthropogenic solar (col3).
Hereby anthropogenic values (col3) were multiplied by 10.

[Figure]

same as above … now for the forcing efficiencies with MACv2

---

## Author Comment (AC1)

We thank both reviewers for their comments and suggestions, which have significantly improved the manuscript. We address each of their comments in turn below. Reviewer comments are in black, and our replies in blue.

**Reviewer 1**

The manuscript addresses the impact of the uncertainties in the aerosol optical properties (aerosol optical depth, single scattering albedo, and asymmetry parameter) on the uncertainty in the surface and Top of the Atmosphere (TOA) direct aerosol radiative effect (DARE) and forcing (DARF). This is done by building and implementing a Monte-Carlo framework. The DARE and DARF uncertainties, despite the progress made in recent years, remain large. The manuscript contributes to the discussion in the scientific community regarding these uncertainties and the results (especially the geographic distributions) are interesting.

However, there is a need for clarifications to be provided, and the discussion should be expanded to include more details, especially in section 3.3. There are also some issues to be addressed, listed below.

l 12 It's better to use the term "aerosol optical properties" instead of "aerosol optical property"

The sentence has been rewritten to be easier to read, and to clarify in addition that both present-day and pre-industrial uncertainties are considered in the study. It now reads, "This framework uses the results of over 2.3 million radiative transfer simulations to calculate global clear-sky DARE and DARF based on a range of uncertainties in present-day and pre-industrial aerosol optical properties, representative of existing and future global observing systems."

Sect. 1. I suggest to provide here more information about the novelty of your study.

We have added a paragraph to the end of the introduction that clarify that the study includes uncertainties in pre-industrial aerosol optical properties and provides a new tool that can translate future instrumental improvements into reductions in aerosol-radiation radiative forcing uncertainty.

l 101: multiplying or dividing?

Multiplying is correct.

l 105: "These are therefore kept constant throughout the whole vertical profile." If I understand well you kept the SSA and g of the total aerosol load constant in the atmospheric column? According to Kinne et al., 2019 "the total values for SSA and g at each altitude depend on the relative AOD contributions. It was not possible to employ this methodology to calculate the vertical profiles of SSA and g? Am I missing something? The lack of knowledge of the vertical distribution of SSA and g induces some biases in DRE calculations (Thorsen et al., 2020)

The optical properties of *each aerosol type* remain constant with altitude, but the contribution of fine and coarse aerosols to total aerosol varies with height in MACv2 (see Table 3 of Kinne et al. (2019)), and therefore when considering all aerosol types in a vertical column, the total aerosol properties do vary with altitude. To clarify that point, the text has been adjusted to read:

"The original MACv2 vertical profiles do not contain information about single scattering albedo or asymmetry factor. These are therefore kept constant throughout the whole vertical profile for each aerosol type. Doing so leads to a vertical variation of the total optical properties of the combined aerosols since the relative proportion of fine and coarse aerosol types varies with height."

ll 130-132: Do you mean "Fig. 2"? Why do you compare the DARE presented in Fig. 1 with the DARF (Fig. 7, Kinne 2019) and not Fig. 5 of Kinne 2019? Maybe you mean Fig. 2 of the manuscript?

Thank you, we meant figure 2. This has been corrected in the manuscript.

l 146: "While there are biases of up to 5% spatially". Do you mean "locally"?

Yes - changed in manuscript.

Section 2.3: To make it more accessible to readers less familiar with Monte-Carlo simulations, it would be beneficial to expand upon it.

A description has been added to clarify the methodology:

"For each given combination of the systematic uncertainties listed in Table 2,500 sets of perturbations to the optical properties are performed in a Monte Carlo framework. First, we draw global offsets to MACv2 AOD, SSA and g by using probability distribution functions that cover the ranges specified in the second column of Table 2 with the shapes specified in the third column. Perturbed distributions are then used in radiative transfer calculations. Finally, these calculations are aggregated to obtain the uncertainty in DARE and DARF."

ll 198-200: It's not clear to me how the employed methodology accounts for covariances between uncertainties in different optical properties.

Yes, the word "covariance" was misleading, since we were not referring to the statistical property but to the fact that the radiative impact of perturbing one optical property depends on which perturbations are applied to the other optical properties. The text has been rewritten to clarify the intended meaning.

ll 204-205: "A draw is taken from a Gaussian distribution centred on the global mean AOD with standard deviation $\sigma_{AOD}$ equal to the AOD uncertainty." So, did you randomly choose an uncertainty from the range 0.005-0.05 (Table 2)? Also, please correct the typo "centred".

Yes, that is indeed the case – we assume that for a given simulation, the uncertainty is determined by a randomly chosen standard deviation taken from the uncertainty distribution. We added a slight clarification to the wording here. We are using British English, thus 'centred' is intended.

l 207: " applied depending on aerosol type". Can you elaborate on this? In the formula below the uncertainty depends on the surface type.

Thank you – we mean depending on surface type. This has been corrected.

ll 218-220: "Since typical ... calculated" This phrase is somewhat confusing. What do you mean by "with perturbations transformed back into $\omega_0$"? You didn't apply perturbations directly to $\omega_0$? Please clarify this.

We agree that this sentence needs further clarification. The distribution was set up in this way since $\omega_0$ is bounded between 0 and 1 but has values typically in the range 0.85-0.95. Given the assumed uncertainty, a Gaussian distribution in $\omega_0$ with a mean in this range will result in randomly drawn perturbations which exceed 1, which would be unphysical. The way around this is to a) assume an uncertainty in $\omega_0$, b) work with co-albedo $(1 - \omega_0)$, and c) use a lognormal distribution in $(1 - \omega_0)$, equivalent to using a normal distribution in $\log(1 - \omega_0)$. A perturbation is drawn from this lognormal distribution, and then we do $1 - (1 - \omega_0)$ to get our resulting perturbation to $\omega_0$.

The manuscript has been updated to explain this better.

l 248: "Figure 5 shows an example of the global-annual mean TOA DARF for one set of input uncertainties". Which specific set?

This has now been added to the text and the figure caption. The set of uncertainties in this case is $\sigma_{AOD}$ = 0.03, $\sigma_{\omega_0}$ = 0.02, $\sigma_g$ = 0.02.

l 263: "shown in Figure 1" Maybe you mean Figure 2 since you are referring to "forcing".

Thank you – yes we mean Figure 2 – this has been changed.

Fig 10 & ll 322-323: I can observe an SSA uncertainty domination at TOA also in the Arctic (albeit not as pronounced as over deserts).

Indeed – this is the case, and the manuscript has been updated accordingly.

l 324-327: "It may also be due to limitations ... for this analysis".

To my opinion, some clarifications are needed here.

Doesn't the larger AOD at longer wavelengths increase the sensitivity of DARE to σAOD over desert regions?

This is true – the larger AOD at longer wavelengths will contribute to both σAOD and σSSA - however the effect of σAOD on DARE uncertainty will be masked partly by the high surface albedo and still remains dominated by σSSA.

Also, you mention that "increased scattering from SSA offset by increased backscatter from asymmetry". Both increased SSA and increased backscatter result in a decrease of the TOA warming/ increase of TOA cooling effect. Why do you use the term "offset"? Also, I would expect that in case of an "offset", the uncertainty of the DARE would decrease and not increase.

We agree that "offset" is not the right word here – we have amended it to "amplified" which more clearly conveys the intent.

l 335: "particularly the strong anthropogenic forcing over East Asia".

Do you mean that "the sensitivities of DARF uncertainty to AOD and SSA are very similar for most regions, particularly over East Asia where there is a strong anthropogenic forcing"? If yes, please rephrase accordingly, as the current sentence is confusing.

Yes this is what we mean. We have adopted a slight variation of the suggested phrasing in the manuscript.

l 337: "This reflects the generally even shape of the contours on Figure" Please elaborate more on this.

We have rephrased this sentence to clarify what we mean, which now reads:

"There are stronger effects from SSA over desert regions and stronger sensitivity to AOD over regions with significant anthropogenic aerosol, such as Southern Africa and North America, as shown by the contours in Figure 8. There is a slightly larger effect globally from AOD, with the contribution from SSA coming next."

l 340: In the phrase "In both cases" it would be better to clarify to which cases you are referring because the current phrasing is somewhat ambiguous

We have clarified that this refers to the surface and TOA DARF cases in the main text.

ll 350-352: "For DARF, the main contributor varies regionally, but AOD generally dominates, except again over bright surfaces, and over remote, low-AOD regions where scattering dominates due to high SSA sea-salt aerosol, where g dominates."There are some issues here:
      1) According to fig. 12a SSA dominates also in central Africa and most of the Indian subcontinent.
      2) g dominates not only over remote regions but also in the region of the Saharan dust outflow
      (tropical North Atlantic).

Please correct the sentence and provide some explanation for the results.

We have added some extra description and explanation of these features to the text.

"The domination of SSA over a larger area in DARF than DARE is due to the presence of absorbing anthropogenic aerosols in these regions, so DARF comes from a relatively larger change in absorption than DARE. The domination of asymmetry in the Saharan dust outflow for DARF is expected from interactions between the DAREs of mineral dust and anthropogenic aerosols: the DARE of mineral dust modifies the radiative fluxes experienced by anthropogenic aerosols for both present and pre-industrial distributions, which amplifies the impact of asymmetry uncertainties."

l 361: When referring to "lookup tables," which lookup tables are you referring to? This is the first time this term is used in the manuscript. Please provide more details.

We have added the following sentence to section 2.3 (Methodology: Uncertainties):

"This process also effectively produces look up tables, consisting of a variety of optical property uncertainties and their associated DARE and DARF uncertainties."

ll 371: "Since B13 only defines uncertainties ... for this case.". It would be useful here to expand the discussion by providing more information about the uncertainties.

The sentence has been clarified, including by giving a clearer source for the uncertainties, "B13 define regional uncertainties for anthropogenic aerosols only (their Table 1), so for consistency the uncertainty in $\sigma_{\omega_0}$ and $\sigma_g$ in each gridpoint is only applied to the anthropogenic part of the total AOD by scaling by the anthropogenic AOD fraction at 550 nm".

ll 384-386: Does the phrase "scaling the values and their associated uncertainties by the global-mean AOD" imply that the AOD-associated uncertainty is linearly proportional to the AOD?

Yes this is what we assume in this specific case. We have added 'assuming a linear dependence between global AOD and its contribution to uncertainty' to confirm this.

ll 397-401: The discussion here is too brief. Please provide more information, especially regarding the non-linearities.

We have re-worded the paragraph somewhat to aid clarity. On reflection, we believe that "non-linearities" is vague and perhaps misleading term. What we aim to say is that there is a degree of additivity that arises when looking at combinations of input uncertainties (e.g. an offset applied to both SSA and AOD), that is not present when looking at the components individually (which is what arises when using the sensitivity coefficients). We find that as a global average, this degree of additivity is around 20%, which suggests that these effects are reasonably significant.

Sect. 4, Table 3: Why did you choose to perform the analysis only using the optical properties uncertainties from Bellouin et al. (2013)?

With the framework we have created, there are an effectively infinite set of choices one could use for the underlying uncertainty distribution of the aerosol optical properties. We use the Bellouin et al. (2013) uncertainties in this section to be consistent with the methodology used in section 3.3, with regionally varying uncertainties, and because Bellouin et al. (2013) use a similar Monte Carlo framework to determine the resulting DARE/DARF uncertainty, allowing for a direct comparison with our framework.

ll 472-474: These are interesting results. To my opinion, they need to be discussed by providing more details (in Sect. 3.3).

We agree that quantifying the change in global DARF uncertainties to a range of reduction in uncertainties, including separating the impact of different regional aerosol uncertainties, would be very informative in terms of determining where instrumental improvements would have the largest payoff. Unfortunately, the project that provided the resources for this work has now ended, so we cannot extend the scope of the study at this time. But the Monte-Carlo framework described in our study will allow us or others to explore those aspects in the future.

References

Kinne, S., 2019. Aerosol radiative effects with MACv2. Atmospheric Chemistry and Physics, 19(16), pp.10919-10959. Doi: 10.5194/acp-19-10919-2019

Thorsen, T.J., Ferrare, R.A., Kato, S. and Winker, D.M., 2020. Aerosol direct radiative effect sensitivity analysis. Journal of Climate, 33(14), pp.6119-6139. doi: 10.1175/JCLI-D-19-0669.1

**Reviewer 2 (Stefan Kinne)**

We thank Stefan Kinne for taking the time to read our manuscript and provide a review.

**Strengths**

Statistical approach (to capture impact uncertainty through multiple simulations)

Simplicity (differences from dual/triple calls)

Link to observational capabilities (sat AOD retrievals, absorption from Aeronet inversions)

**Weaknesses**

Implications to climate prediction uncertainty are overdrawn (without indirect effects)

The implications that we state for our study are reasonable and are clearly placed in the broader aerosol effective radiative forcing context. Our study focuses on direct effects (aerosol-radiation interactions) because their uncertainty remains sizeable, but we fully acknowledge in the introduction the importance of indirect effects (aerosol-cloud interactions) and the total aerosol effective radiative forcing for climate prediction. As we note in the introduction, direct and indirect effects are controlled by different aerosol and environmental properties, which justifies looking in detail at one of them, as done in this study.

All-sky impacts (clear-sky to all-sky) are based on a simple approximation

We agree that our inclusion of all-sky effects is relatively simple. This is intended as an illustration of the impacts of clouds. It is however plausible that the contribution of clear-sky forcing uncertainty to all-sky forcing uncertainty does in fact scale with the clear-sky fraction, with additional uncertainty being contributed by cloudy-sky forcing but not quantified in our study. We added "Applying a simple scaling to all-sky conditions…" to the abstract to further clarify this. We also added the following text to the conclusion to reflect this method:

"Finally, our estimation of the all-sky DARE and DARF is based on a simple scaling based on cloud optical depth and cloud fraction, as described in Section 4. More complex methods could be applied and could form the basis of further work. However, the simple method used here provides a first-order estimate of the contribution of clear-sky DARF uncertainty to all-sky DARF uncertainty."

Pre-industrial aerosol properties are not (but should be) considered in DARF

Pre-industrial aerosol properties and their uncertainties are considered in our calculations of both DARE and DARF. This is one of the distinguishing features of our study.

We added "as well as pre-industrial uncertainties" to the abstract to clarify this. Additionally, the following text in the methodology clearly states that pre-industrial aerosol properties are included:

"Aerosols are prescribed using the MACv2 aerosol climatology (Kinne, 2019a). MACv2 provides AOD, SSA (denoted $\omega\_0$), and g for each month of the year for both present-day and pre-industrial cases for different aerosol types."

When considering DARF we ensure that the uncertainties are drawn from the same distribution for the pre-industrial and present-day calculations to avoid biasing uncertainty in DARF (which comes from the uncertainties in the anthropogenic aerosol optical properties).

DARE uncertainties do not include longwave effects (thus likely could be too large)

Agreed – we explicitly only look at the shortwave effects here. We have added 'shortwave' into the title and at three more points in the abstract to make this even more explicit. We already include a discussion of the LW effects in the conclusion, where we discuss how the LW effects are most important for coarse aerosols, and thus particularly for mineral dust (i.e. natural aerosols) and therefore expected to impact the DARE rather than the DARF. It is however unclear whether including LW effects would compensate for some of the SW uncertainties.

The paper applies aerosol radiative properties of the MACv2 aerosol climatology as a base-line for uncertainties of aerosol radiative effects and aerosol forcing – but only for solar impacts under cloud-free conditions (at TOA and surface) although with an approximation to estimate all-sky impacts. This study is interesting, but there are also aspects that need to be clarified and possibly corrected during the review process.

While MACv2 aerosol properties are applied to define the solar spectral aerosol properties (AOD, SSA, ASY) and the AOD vertical distribution, questions remain as to why the calculated results are so different from 'official' MACv2 simulations. For clear sky solar (all) aerosol radiative at the surface the global average -8.6 W/m2 cooling is larger than -7.4 W/m2 in MACv2 (and -7.2 W/m2 in MACv3). This suggests that applied solar AOD values of the implementation in this study are larger. This then could also explain – with a similar absorption potential - why TOA impacts at -4.5W/m2 are more negative (than -3.5W/m2 in MACv2 and -3.7W/m2 in MACv3). It would be very useful to compare global 'input' AOD maps (especially for solar spectral AOD data) to identify the cause for these differences - under the premise that an identical input to aerosol properties was used. Suggested contributions of (environmental) non-aerosol properties are weak, because DARF and DART estimates are based on differences from two simulations, where these properties do not vary (just largely cancel in their impacts).

It is unclear why results between MACv2 and our study disagree to the extent shown in Table 1, and unfortunately the project that supported the study has ended, limiting our capability to investigate differences further.

The estimated aerosol property uncertainty impact on DARE ignored longwave effects, which possibly could reduce uncertainties. For climate relevant DARF (at TOA) the solar impact is sufficient, but other important elements (the pre-industrial uncertainty and more important indirect effect through clouds) are not included, so that climate uncertainty relevance (for me) is more than limited.

As stated in our response to Stefan Kinne's summary comment above, we fully acknowledge that our study has a specific focus, and that many other aspects of aerosol radiative forcing (longwave, cloudy-sky, aerosol-cloud interactions) are important. Indeed, much of our research activities beyond this study addresses those additional aspects. The introduction and conclusion acknowledge those points clearly. But within its specific scope, the study brings new insights into the global and regional structure of DARE and DARF uncertainties and proposes a way to propagate future reductions in AOD, SSA, and asymmetry uncertainties to DARE and DARF. For these reasons, the study is relevant to the much broader effort of reducing climate uncertainties.

Finally the applied observational uncertainties should be reviewed. For AOD I would include an uncertainty that increase also with AOD and for SSA I would ties uncertainty directly to AOD … possibly by using AAOD (AOD*[1-SSA]) uncertainty instead.

We agree that observational uncertainties depend on AOD, but in a complex way that the remote sensing community is only beginning to understand. From the viewpoint of AERONET sunphotometer retrievals, which is that taken in our study, AOD uncertainties depend on the conditions of the retrievals (cloud masking, solar zenith angle, aerosol shape), so their dependence on the AOD level is complex. AERONET has begun only recently to quantify and publish uncertainties of individual retrievals, so it will be interesting to assess whether AOD-dependent patterns emerge. For SSA, uncertainties decrease in a power law with increasing AOD (Sinyuk et al., 2020 https://doi.org/10.5194/amt-13-3375-2020), which, if confirmed, would be interesting to include in a follow-up study.

To acknowledge these elements, we have added the following to the conclusion:

"We assume that AOD and SSA uncertainties are independent of AOD. The strength of that assumption is difficult to assess for AOD, because uncertainties in individual AERONET AOD measurements depend on errors due to cloud masking, viewing geometry, and assumptions on aerosol shape that are AOD independent. It is unclear how those uncertainties average into an AOD-dependent behaviour. The situation is much clearer with SSA uncertainties and Sinyuk et al. (2020) suggest that SSA uncertainties decrease in a power law with increasing AOD, suggesting the high tail of our DARE uncertainty distribution is overestimated."

Overall, it is an interesting sensitivity study.

Thank you for the positive comments, and for taking the time to read and provide constructive feedback.